# Free choice shapes normalized value signals in medial orbitofrontal cortex

Hiroshi Yamada[1,2,3,4], Kenway Louie[1], Agnieszka Tymula[1,5] & Paul W. Glimcher[1,6]

Normalization is a common cortical computation widely observed in sensory perception, but its importance in perception of reward value and decision making remains largely unknown. We examined (1) whether normalized value signals occur in the orbitofrontal cortex (OFC) and (2) whether changes in behavioral task context influence the normalized representation of value. We record medial OFC (mOFC) single neuron activity in awake-behaving monkeys during a reward-guided lottery task. mOFC neurons signal the relative values of options via a divisive normalization function when animals freely choose between alternatives. The normalization model, however, performed poorly in a variant of the task where only one of the two possible choice options yields a reward and the other was certain not to yield a reward (so called: "forced choice"). The existence of such context-specific value normalization may suggest that the mOFC contributes valuation signals critical for economic decision making when meaningful alternative options are available.

[1] Center for Neural Science, New York University, 4 Washington Place, Room 809, New York, New York 10003, USA. [2] Division of Biomedical Science, Faculty of Medicine, University of Tsukuba, 1-1-1 Tenno-dai, Tsukuba, Ibaraki 305-8577, Japan. [3] Graduate School of Comprehensive Human Sciences, University of Tsukuba, 1-1-1 Tenno-dai, Tsukuba, Ibaraki 305-8577, Japan. [4] Transborder Medical Research Center, University of Tsukuba, 1-1-1 Tenno-dai, Tsukuba, Ibaraki 305-8577, Japan. [5] School of Economics, University of Sydney, Room 370, Merewether Building (H04), Sydney, New South Wales 2006, Australia. [6] Institute for the Interdisciplinary Study of Decision Making, New York University, 300 Cadman Plaza West, Suite 702, Brooklyn, New York 11201, USA. Correspondence and requests for materials should be addressed to H.Y. (email: h-yamada@md.tsukuba.ac.jp)

A growing body of evidence indicates that value signals distributed in the brain shape decision-making behavior[1–3]. Such value signals are especially prominent in the orbital and medial areas of prefrontal cortex[4] and the parietal cortex[5,6] where neural activity represents value information in a diverse array of paradigms[7]. Notably, these value signals do not simply reflect the fixed values assumed by many models of choice[8–10], but instead the magnitudes of these value signals have been shown to depend on present or past alternatives[11–15]. For example, a pioneering finding in orbitofrontal cortex (OFC) indicates that OFC neurons signal the relative values of food items among the alternatives monkeys have recently encountered in a block of trials[16]. This finding implies that value signals identified in the OFC may reflect comparative computations such as "divisive normalization", a common cortical computation for relative information coding proposed to explain nonlinear response properties in sensory cortices[17]. However, it remains unclear how or if the value signals in these prefrontal areas are normalized and incorporated into the process of choosing among alternatives.

To investigate the direct link between normalized values signals and choice behavior, we focused on the medial orbitofrontal cortex (mOFC, see Rudebeck and Murray)[4,7]. mOFC is a subdivision of the OFC medial to the medial orbital sulcus (Brodmann's area 14, 13a, 13b, and 11m), and reciprocally connected to both medial and orbital prefrontal network areas. Although previous studies have identified neural signals related to reward values in the OFC, they have not specifically searched for normalized value representations in prefrontal areas. For example, human ventromedial prefrontal cortex (vmPFC), mostly along the medial wall, has been shown to represent a diverse set of reward values in various behavioral tasks, including both active value-guided decision making[18–22] and passive item valuation[23,24] when no choice is made. Single neuron activity in monkey vmPFC carries value signals that reflect offer values of gambles[25], motivational level[26,27] and a possibility of reward[28]. In the lateral subdivision of OFC (lOFC, a subdivision of OFC lateral to medial orbital sulcus), neurons have been shown to signal the relative values of items when monkeys perform behavioral tasks both with and without choices[11,12]. Value signals are evident across all of these prefrontal network areas; however, none of the areas has been examined to determine whether these value signals employ a computational process, divisive normalization, when animals choose freely among items of different reward values.

We thus specifically targeted the mOFC to test whether single mOFC neurons signal the normalized values of rewards when monkeys made "free choices": choices between two available rewarding items. We found that a common cortical computation, divisive normalization, is implemented in the activity of mOFC neurons representing reward values under these conditions. These normalized value signals were prominent when monkeys made free choices, but surprisingly were attenuated when monkeys were "forced" to choose one of the options: when one of the two possible rewards was signaled to have zero value or impact with certainty and the other was potentially rewarding, a situation colloquially referred to in the neuroscience literature as a "forced choices" (a nomenclature we adopt in this paper)[29].

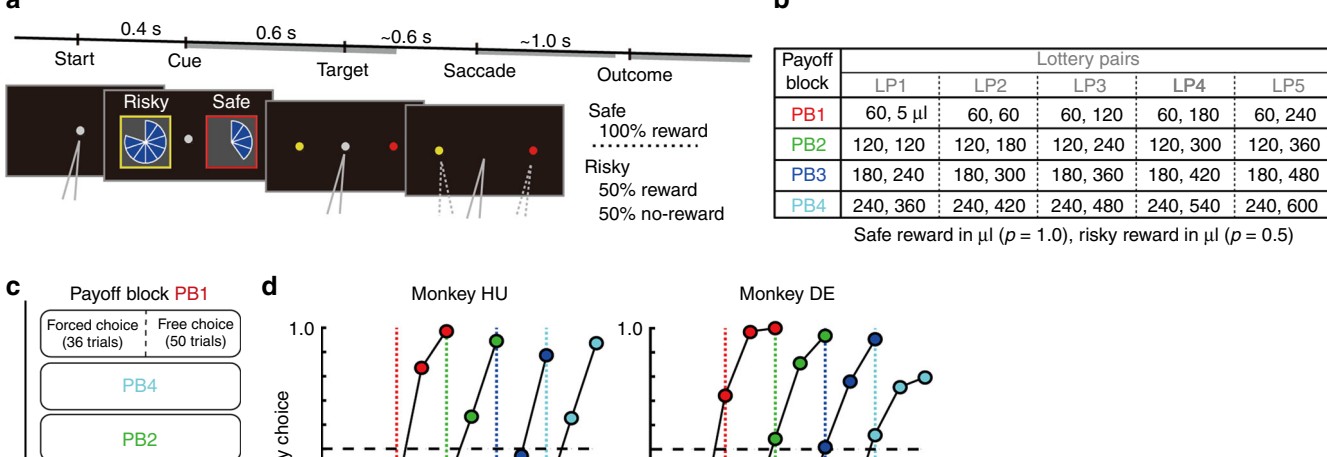

**Fig. 1** Lottery task and choice behavior. **a** A sequence of events in free choice trials. Pie charts indicated reward magnitudes from 60 to 600 µl in 60 µl increments. Gray color of the central fixation target indicated that the monkeys could choose either option freely. In the forced choice trials (red or yellow fixation color), monkeys were required to choose the color-matched target among the alternatives, unless otherwise the trials were aborted. Positions of the risky and safe options were fixed during a single payoff block. Gray bars (top) indicate the 1.0 s time periods used to analyze neuronal activity; cue, saccade and feedback periods. **b** Payoff matrix: in each payoff block 1 to 4, the monkeys chose between a 100% fixed amount of water reward and a lottery that would deliver a reward with 50% probability (5 different risky reward magnitudes per one block). For example, in payoff block 1 (PB1), the safe 60 µl reward was represented by a 1/10 filled pie chart and the risky option was represented by a pie chart ranging from empty to 4/10 full. **c** An example payoff block sequence (randomly selected without replacement until all four payoffs were presented). In a block the first 36 trials were forced choice trials. Then, 50 free choice trials (10 of each type) followed in random order. **d** Percentages (P) of risky choice plotted against magnitude of risky reward in each PB (indicated by color). Dashed colored lines indicate where risky and safe options have equal expected value

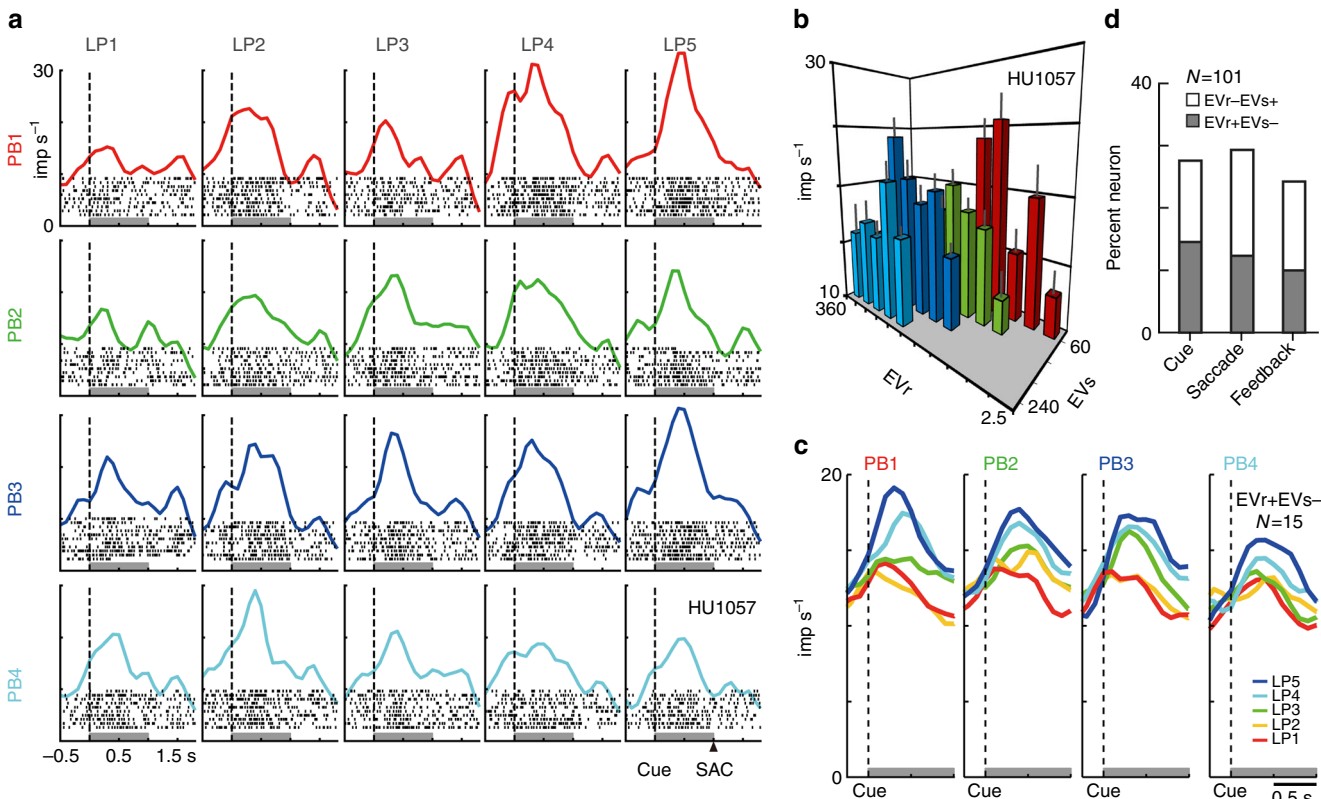

**Fig. 2** Relative value signals in the activity of mOFC neurons. **a** Rasters and histograms of an example mOFC neuron modulated by the relative value of options. The activity aligned at cue onset during free choice trials was shown for 20 lottery pairs (four PBs times five LPs, 200 trials). Black dots in the histograms indicate raster of spikes. Gray bars indicate the cue period to estimate the neuronal firing rates shown in **b**. SAC indicate approximate time of saccade onset. **b** Activity plot of the mOFC neuron in **a** against the expected values of risky (EVr) and safe option (EVs). Error bars indicate s.e.m. The neuron showed positive and negative regression coefficients for EVr and EVs (EVr+EVs− type, EVr, 0.042, EVs, −0.048, AIC = 1283), respectively. **c** Activity histogram of 15 mOFC neurons modulated by relative values of risk and safe options during cue period (EVr+EVs− type). Activity in each of four payoff blocks (PB1–4) is shown for the five types of lottery pairs (LP1–5). **d** Percentage of mOFC neurons modulated by relative values during three task periods. Gray indicates activity showing the positive and negative regression coefficients for EVr and EVs, respectively (EVr+EVs− type). White indicates activity showing negative and positive regression coefficients for EVr and EVs, respectively (EVr−EVs+ type)

## Results

**Cued-lottery task in monkeys.** To examine value coding during economic choice behavior, we trained two monkeys to perform a cued-lottery task with varying reward payouts and probabilities (Fig. 1). During the task, visually displayed pie charts indicated reward magnitudes to the monkeys, while risky (50% reward, otherwise nothing) and safe (100% reward) options were presented on the left and right side of fixation in each block of trials (Fig. 1a). Monkeys made choices between the risky and safe options among 20 lottery pairs (Fig. 1b); these pairs were divided into four separate groups of lottery pairs (five risky options against one safe option) and presented to the monkeys as blocks of trials (Fig. 1c, payoff block (PB)). In each block 36 "forced choice" trials were followed by 50 "free choice" trials. A gray central fixation stimulus indicated free choice trials, while a red or yellow central fixation stimulus indicated forced choice trials in which only a choice of the color-matched target would yield a reward. In each PB, the five lottery pairs were systematically matched in terms of their relative values with the expected value of risky option (Fig. 1b, LP1–5): considerably larger than the safe option (LP5); slightly larger (LP4); equal (LP3); slightly smaller (LP2); or considerably smaller (LP1). Together, these four blocks allowed us to examine the extent of relative value coding in mOFC neurons.

Details of the behavioral training, learning progress and behavioral performance of the animals in the lottery task have

been reported previously[30]. Briefly, after completing the training, monkeys learned the expected values of risky and safe options, and chose risky options more frequently if the expected values of risky options were higher than those of safe options and vice versa (Fig. 1d). Behavioral measures, such as percent correct trials and saccade reaction time, were not consistently related to expected value between monkeys (Supplementary Fig. 1), suggesting that potential confounding factors such as motivation or attention did not vary between conditions. To examine the mechanism by which mOFC neurons signal values, we sampled 182 mOFC neurons (Supplementary Fig. 2). Of these sampled units, 101 neurons (50 and 51 neurons from monkey DE and HU, respectively) were recorded and analyzed during all or almost all of the four PBs while monkeys were engaged in the lottery task (minimum 200 trials).

**Relative value coding in mOFC neurons.** We first examined whether the activity of mOFC neurons represents relative value information in a general way (without utilizing normalization equations specifically in our analysis; see Methods), as has been seen in an adjacent area, the lOFC[16], where neurons have been shown to signal the relative values of items among possible alternatives in a block of trials. Cue period activity from an example relative value coding neuron from our dataset is shown in Fig. 2a. In each payoff block differentiated by color, the neuron

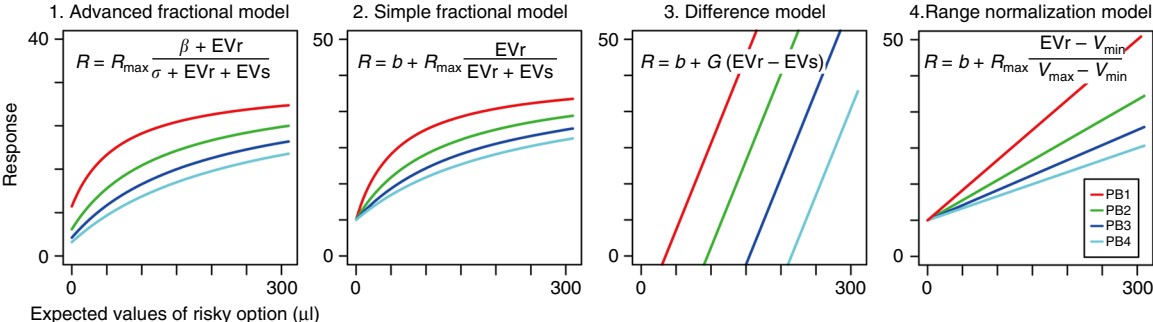

**Fig. 3** Potential normalized value coding models. Schematic depiction of predicted neuronal responses in the four alternative normalized value coding models. In each panel, four colored lines indicated the model output (y-axis) in each of payoff block (PB1–4) plotted against the expected values of risky option (x-axis). Expected values of safe option were 60, 120, 180 and 240 µl in PB1 to 4, respectively. Model equations are shown in each plot. $R_{max}$, $\beta$, $\sigma$, $b$ and $G$ were free parameters. For this schematic drawing, the following values for free parameters were used; 1. Advanced fractional model, $R_{max}$, $\beta$ and $\sigma$ were 40 spk s$^{-1}$, 20 and 10 µl, respectively; 2. Simple fractional model, $R_{max}$ and $b$ were 40 and 10 spk s$^{-1}$, respectively; 3. Difference model, $G$ and $b$ were 0.4 (a.u.) and 10 spk s$^{-1}$, respectively; 4. Range normalization model, $R_{max}$ and $b$ were 40 and 10 spk s$^{-1}$, respectively. See Methods for more details

showed increasing activity as the relative value of risky options increased (LP1 to 5): the larger the expected value of the risky option compared to the safe option, the higher the neural activity. This activity modulation diminished as the expected value of the safe option increased from PB1 to PB4. Consistent with a relative value representation, the activity of this neuron was modulated by the expected value of both the risky (EVr) and safe (EVs) options, with opposite modulation effects (Fig. 2b, $n = 200$, Akaike's information criterion (AIC) = 1283, regression coefficient; EVr, 0.042, $P < 0.001$; EVs, −0.048, $P < 0.001$; intercept, 19.6, $P < 0.001$). This relative value coding was found in 28% of mOFC neurons during the cue period. Of the mOFC neurons, 15% (15/101) showed increasing activity as the expected values of risky option increased and of safe options decreased (Fig. 2c, EVr+EVs type), while 13% of neurons (13/101) showed increasing activity as the expected values of risky options decreased and of safe options increased (EVr−EVs+ type). Relative value signals of this kind were evident across the entire decision-making interval (Fig. 2d): when monkeys made decisions based on cue information (cue period, 28%), after saccadic decisions and prior to outcome feedback (saccade period, 29/101, 29%), and during outcome feedback (feedback period, 24/101, 24%); see gray lines in Fig. 1a for three task periods: cue period (1.0 s window after cue onset), saccade period (1.0 s window after saccade onset) and feedback period (1.0 s window after feedback onset). There was no significant difference in the proportion of modulated neurons among the task periods ($\chi^2$ test, $n = 303$, $P = 0.584$, $\chi^2 = 1.075$, df = 2). In total, 27% (81/303) of the task periods showed activity modulation by the relative value of options, and 48 neurons exhibited relative value coding in at least one of the three task epochs. These 81 relative value signals were used in further analyses to test in greater detail how the value signals are normalized. Note that only a small percentage of neurons exclusively encoded choice location (7/101, 7/101 and, 5/101 during cue, saccade and feedback periods), consistent with previous findings in lOFC[16,31].

**Normalized value coding in mOFC neurons**. A common cortical computation underlying relative information coding in both sensory and decision-making brain regions is divisive normalization[13,17]. To examine the role of divisive normalization in mOFC relative value coding, we fit the observed mOFC data to a standard normalization equation:

$$R = R_{max} \frac{\beta + EV_1}{\sigma + EV_1 + EV_2}$$

where the firing rate $R$ depends on the expected values of both alternatives. For a given neuron, $EV_1$ and $EV_2$ were the expected values of the two options. If a neuron increased firing rate as the value of the risky option increased, then $EV_1$ was defined as the risky option and $EV_2$ as the safe option. If a neuron increased firing rate as the safe option increased in value, then $EV_1$ was defined as the safe option and $EV_2$ as the risky option. $R_{max}$, $\beta$ and $\sigma$ were free parameters, with $R_{max}$ characterizing the maximal level of neural activity. $\beta$ and $\sigma$ determine the contribution of the expected values to neuronal responses, with $\beta$ governing the level of activity at "baseline" (when both $EV_1$ and $EV_2$ are zero) and $\sigma$ determining the sensitivity of neuronal responses to the expected values (large $\sigma$ means low sensitivity). We refer to this common normalization equation as the "advanced fractional model", and note that it yields non-linear responses to changes in the expected values as shown in the output response visualized in Fig. 3 (left panel, advanced fractional model).

We first fit the advanced fractional model to the activity of mOFC neurons during "free choice" trials (trials on which both the risky and safe options offered non-zero expected values), and compared this advanced fractional model (M1) with other possible functional forms of normalization: a "simple" fractional model (M2), a difference model (M3) that has been argued for in some cortical[32] and subcortical structures[33] and a range normalization model (M4) previously used in studies conducted in the lOFC[11,12] (see Fig. 3 and Methods for details). To determine which model best describes observed mOFC activity, we compared the AIC term for each model. AIC measures the goodness of model fit with a penalty for the number of free parameters employed by the model. As demonstrated for an example neuron (Fig. 4a; same neuron as Fig. 2a), the advanced fractional model was the best-fitting model among the four alternatives we explored ($n = 200$, see AIC values in Fig. 4a, percent variance explained: trial-based, 13.5%; mean responses-based, 46%). For each neuron and task epoch with relative coding activity, we quantified AIC differences between alternative models and determined which model showed the smallest AIC values across the population. These AIC differences indicated that the advanced fractional model best for described mOFC activity at the population level (Fig. 4b, $n = 81$, one-sample t-test, df = 80; M1–M2, $P < 0.001$, $t = −4.35$; M1–M3, $P < 0.001$ $t = −3.53$; M1–M4, $P < 0.001$, $t = −4.10$). We also confirmed that the advanced fractional model was better than other potential alternative models, including ones representing the expected values of risky options, expected values of safe options, expected values of chosen options and the choice of risky options, as well

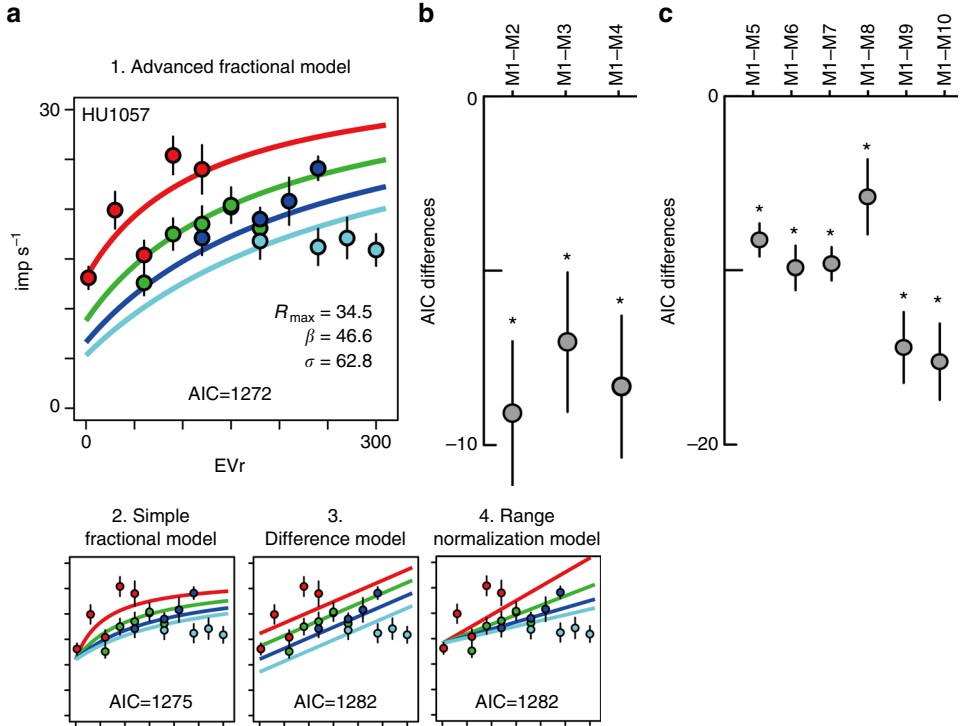

**Fig. 4** The advanced fractional normalization model best explained mOFC relative value coding. **a** Four model outputs fit to the example neuronal activity encoding relative value (same neuron as shown in Fig. 2a). Average firing rates and s.e.m. in 20 lottery pairs are plotted in each panel. Colored lines indicate the best-fit lines segregated by payoff block. **b** Plots of the AIC differences between models estimated across the population. Mean and s.e.m. were estimated for 81 activities that showed relative value coding. AIC differences between model 1 and the other three relative expected value models are shown. **c** Same as **b**, but for AIC differences between model 1 and alternative models 5–10. See Methods for details of the models. Asterisk indicates statistical significance of the AIC differences from zero at $P < 0.01$ using one sample $t$-test

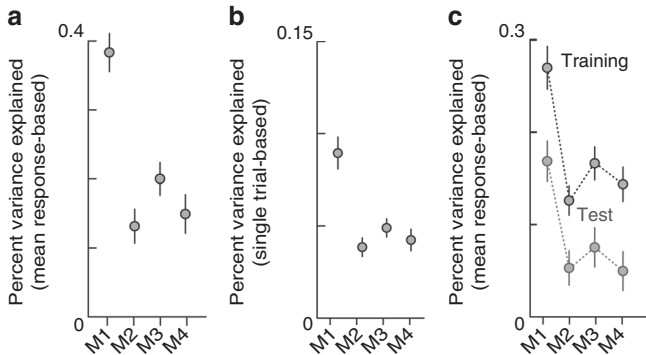

**Fig. 5** Comparisons of the model performances for relative value coding. **a** Plots of the percent variance explained by the four normalization model for the mean response-based data estimated in 20 lottery pairs. **b** Same as **a**, but for the single trial-based data. **c** Plots of the percent variance explained by the models for the mean response-based data when cross-validation was performed. Percent variance explained for training data and test data are shown

as a null model and an artificial model (Fig. 4c, $n = 81$, one-sample $t$-test, df = 80; M1–M5, $P < 0.001$, $t = -8.71$; M1–M6, $P < 0.001$, $t = -7.76$; M1–M7, $P < 0.001$, $t = -10.2$; M1–M8, $P = 0.009$, $t = -2.68$; M1–M9, $P < 0.001$, $t = -7.09$; M1–M10, $P < 0.001$, $t = -6.96$). In summary, of the models tested, relative value coding in the activity of mOFC neurons was most consistent with a divisive normalization computation.

To evaluate the performance of the model, we estimated percentages of variance explained (see Methods). The divisive

normalization model performed well compared to the other three relative value models (Fig. 5), as 40% of the variance was explained by the advanced fractional model in the mean response-based estimation in 20 lottery pairs (Fig. 5a, $n = 81$, paired $t$-test, df = 80; M1 vs. M2, $P < 0.001$, $t = 8.38$; M1 vs. M3 EVs, $P < 0.001$, $t = 6.54$; M1 vs. M4, $P < 0.001$, $t = 7.65$). Similar results were obtained when the percent variance explained was estimated based on single trial data (Fig. 5b, $n = 81$, paired $t$-test, df = 80; M1 vs. M2, $P < 0.001$, $t = 5.87$; M1 vs. M3 EVs, $P < 0.001$, $t = 4.94$; M1 vs. M4, $P < 0.001$, $t = 5.55$), though as expected the single trial-based percent variance explained was lower than the mean response-based measure due to trial by trial variability in the neural activity. Furthermore, cross-validation demonstrated model explanatory power in test data as well as training data, with the advanced fractional model remaining the best model (Fig. 5c, test data: $n = 81$, paired $t$-test, df = 80; M1 vs. M2, $P < 0.001$, $t = 5.39$; M1 vs. M3 EVs, $P < 0.001$, $t = 4.55$; M1 vs. M4, $P < 0.001$, $t = 5.45$). Note that percent variance explained decreased even in the training data since the data size was half the size of the full data set.

To examine the descriptive ability of the advanced fractional model, we verified whether the estimated normalization parameters appropriately described all aspects of neural activity. Across our population, estimated parameters were stable and within reasonable ranges, with an $R_{max}$ of ~20 imp s$^{-1}$ (Fig. 6a, $n = 81$, Kruskal–Wallis test, $P = 0.44$, $H = 1.62$, df = 2), a $\beta$ of ~80 µl (Fig. 6b, $P = 0.16$, $H = 3.72$, df = 2) and $\sigma$ of ~90 µl (Fig. 6c, $P = 0.07$, $H = 5.38$, df = 2). Notably, estimated $R_{max}$ values were strongly correlated with observed maximal firing rates (Fig. 6d, $n = 81$, $r = 0.68$, $P < 0.001$, $t = 8.18$, df = 79). Estimated $\beta$ and $\sigma$ parameters were also reliable as follows. We quantified $R_{max} \beta \sigma^{-1}$, a term equivalent to output of the normalization equation when $EV_1 = EV_2 = 0$; this quantity can be thought of as

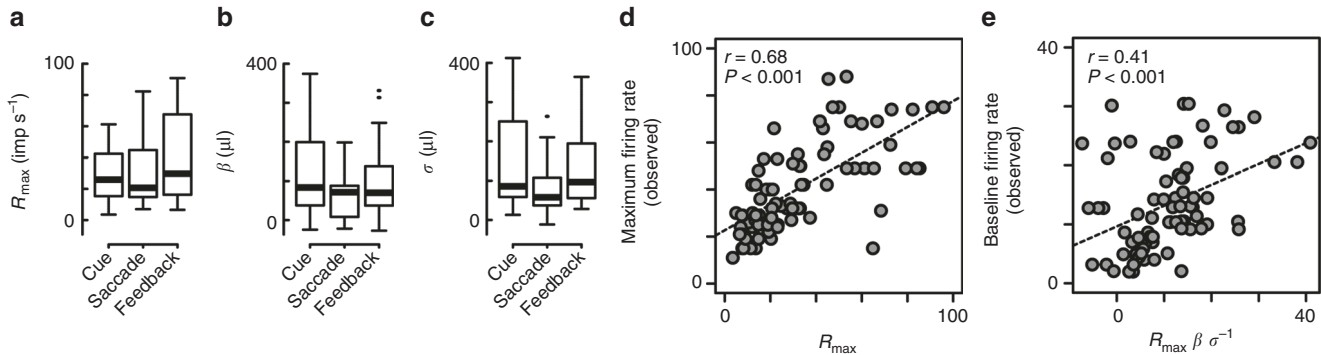

**Fig. 6** Comparison of the estimated normalization parameters and observed firing rates. **a–c** Box plots of the estimated parameters in the advanced fractional model. The $R_{max}$, $\beta$, and $\sigma$ were plotted separately during three task periods. **d** Plots of the maximal firing rate observed in each mOFC neurons against the estimated $R_{max}$. **e** Plots of the baseline firing rate observed in each mOFC neurons against the model output with no value information ($R_{max}\,\beta\,\sigma^{-1}$). Dashed lines in **d**, **e** indicate regression slopes. Correlation coefficients and statistical significance are shown

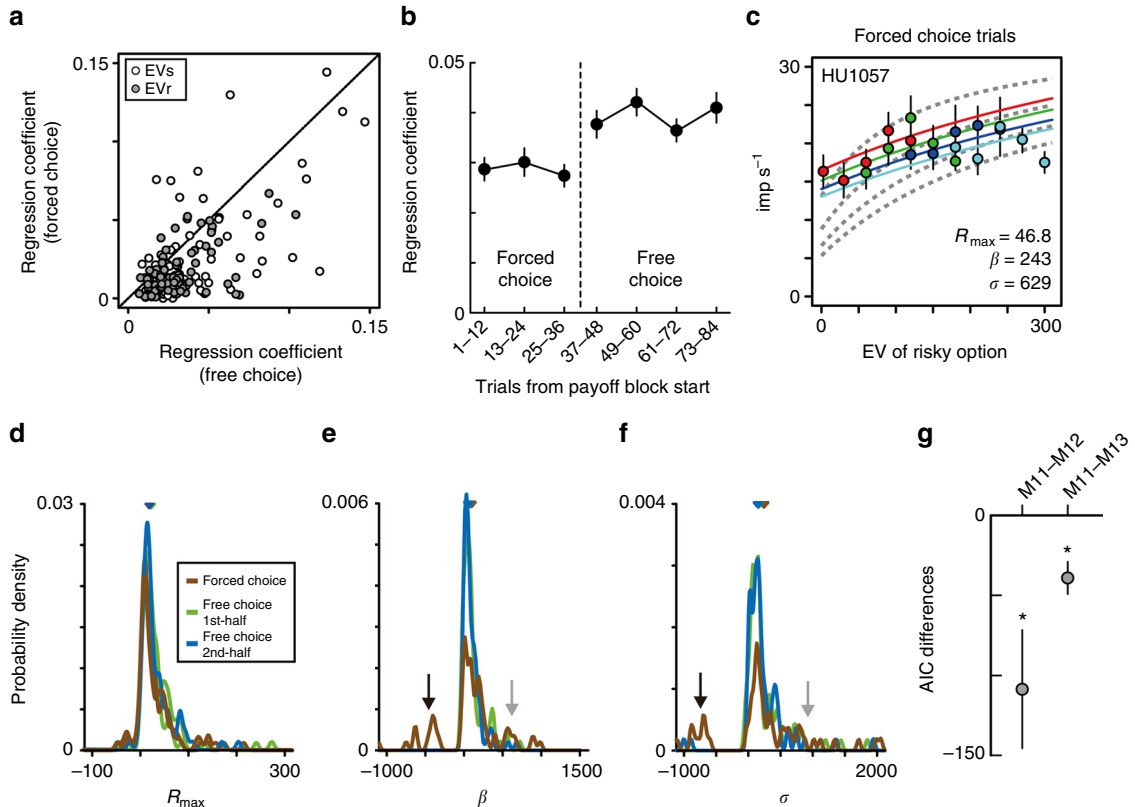

**Fig. 7** Attenuated value coding of mOFC neurons during forced choice trials. **a** Plots of the absolute value of regression coefficients for EVr (gray) and EVs (white) during free and forced choice trials. Mean ± s.e.m. during free and forced choice trials: EVr, 0.031 ± 0.002, free choice, 0.017 ± 0.002, forced choice; EVs, 0.042 ± 0.003, free choice, 0.027 ± 0.003, forced choice. **b** Average of the absolute value of regression coefficients for EVr and EVs across the trial block. Regression coefficients were estimated every 12 trials from the start of the payoff block. Error bars indicate s.e.m. **c** Activity plots of the same neuron in Fig. 4 during the forced choice trials. Color lines indicated the best-fit lines during the forced choice trials. Gray lines indicated the best-fit lines during the free choice trials as shown in Fig. 4a. **d–f** Probability density of the estimated parameters of the models during forced choice trials (brown), the 1st half of the free choice trials (green), and 2nd half of the free choice trials (blue). In **d–f**, triangles in the figures indicate the median. **g** Plots of the AIC differences between models estimated across the population. AIC differences between model 11 and models 12–13 are shown. Error bars indicate s.e.m. In **a–g**, the results during forced choice trials were shown when assuming expected values of risky and safe options were defined as indicated by pie chart stimuli (assumption 1, see Methods for details)

representing baseline firing rates in the normalization model[34]. Across our population, $R_{max}\,\beta\,\sigma^{-1}$ values were significantly correlated with observed baseline firing rates before the cue stimuli appeared (Fig. 6e, $n = 81$, $r = 0.41$, $P < 0.001$, $t = 4.00$, df = 79). Thus, the estimated parameters of the normalization model appear to appropriately capture aspects of the observed neural activity, suggesting that the advanced fractional model may underlie relative value signals in mOFC neurons.

**Decision context and normalized value signals in mOFC.** To further test whether a normalized value code is specifically related

to decision making during free choice, we examined neural activity during the "forced choice" trials presented to the monkeys at the beginning of each payoff block. These forced choice trials presented identical task timing, cue displays and reward contingencies as the previously described free choice trials; however, in the forced choice trials, the fixation target color (red or yellow) instructed the monkeys that only the color-matched target would yield a reward (and that the other target was certain not to provide a reward).

When the monkeys were instructed by the computer to "choose," the relative value signals evident in the regression coefficients for the expected values of risky and safe options were weak when compared to those observed on free choice trials in the activity of the same neurons (Fig. 7a, $n = 81$, paired $t$-test; EVr, $P < 0.001$, $t = 7.67$, df = 80; EVs, $P < 0.001$, $t = 4.98$ df = 80; see also Supplementary Fig. 3 for activity histogram). While the forced choice trials were presented to the monkeys at the beginning of PBs, weak modulation in the forced choice trials was not due to an adaptation process, as might be postulated to occur in adjacent area lOFC[11,12]. The weak modulation in the forced choice trials were maintained throughout forced choice trials (Fig. 7b, one-way analysis of variance (ANOVA): forced choice trials, $n = 486$ $(81 \times 3 \times 2)$, $P = 0.75$, $F = 0.29$, df = (2, 483)). Stronger modulation appeared only after the start of free choice trials (paired $t$-test, $P < 0.001$, $t = 3.66$, df = 161, the last 12 forced choice trials vs. the first 12 free choice trials) and was maintained through a payoff block (one-way ANOVA: free choice trials, $n = 648$ $(81 \times 4 \times 2)$, $P = 0.35$, $F = 1.09$, df = (3, 644)). Thus, relative value coding in mOFC neurons was apparently weaker when monkeys were forced to choose one option.

Next, we examined the computational basis of these effects by fitting the advanced fractional model to neuronal activity during forced choice trials. Note that mOFC neurons could encode the expected values of risky and safe option in two possible ways: their activity could reflect the non-selectable option having the value indicated by the pie chart stimulus (as in the free choice trials) or the non-selectable option having a value of zero (we tested both of these possibilities, see Methods). The model fit to forced choice data in an example neuron (same neuron as in Fig. 4a) showed an attenuation of the activity modulation by relative value, evident as increases in both $\beta$ and $\sigma$ (Fig. 7c, $\beta$ increased from 47 to 243 µl; $\sigma$ increased from 63 to 629 µl), with a slight increase of $R_{max}$ from 35 to 47 Hz. The increase in $\beta$ and $\sigma$ parameters produces a decreased sensitivity to relative value information, which is evident as a shallower slope of model responses during forced choice trials (color lines) compared to free choice trials (gray dashed lines). Across our population, we found increases in estimated $\beta$ and $\sigma$ parameters in forced choice trials in several cases (Fig. 7e, f, see brown line indicated by gray arrows), but also occasional negative values (indicated by black arrows). In contrast to the similar distribution between early (green) and late (blue) free choice trials, the parameter distributions became wider and the density of the peak values decreased during forced choice trials (brown) ($n = 243$ $(81 \times 3)$, Brown–Forsythe test: $\beta$, $P = 0.022$, $F = 3.89$, df = 241; $\sigma$, $P < 0.001$, $F = 22.4$, df = 241). The distribution of $R_{max}$ parameters during forced choice trials was also changed (Fig. 7d, $P < 0.001$, $F = 16.15$, df = 241). Negative values in estimated $\beta$ and $\sigma$ indicated that the advanced fractional model was no longer well fit to the weak value modulations observed in some neuronal activity. Indeed, performance of the model in describing neuronal activity was worse in forced choice trials than in the free choice trials (Supplementary Fig. 4). Among the four tested models, however, the advanced fractional model remained the model that best characterized mOFC activity in the forced choice trials (Supplementary Fig. 5). In addition, the activity difference

between free and forced choice trials was not better explained by behavioral measures, such as percent correct trials or saccadic reaction times (Fig. 7g, $n = 81$, one-sample $t$-test, df = 80; M11–M12, $P = 0.004$, $t = -2.94$; M11–M13, $P < 0.001$, $t = -3.87$). Thus, the task context for value-based decision making—free versus forced choice—changes the normalization computation in mOFC neurons.

**mOFC normalized value signals and risk attitudes of monkeys.** Lastly, we examined whether the divisively normalized value signals observed in mOFC activity were related to other aspects of the decision-making process, in particular the risk attitudes of the monkeys. We estimated the correlation coefficient between behavioral risk attitudes (percentages of risky choice when a neuron was recorded) and neuronal activity, examined in trials where the expected values of safe and risky option were identical. Specifically, we examined whether firing rates in the equal expected values trials were consistently deviated from the mean firing rates of the neuron according to the monkey's risk attitude; under a subjective value code, neural activity would be systematically deviated from a linear function as a function of risk preference of monkeys. We found a slight correlation between neuronal activity and percentages of risky choices with opposite signs of the effects among EVr+EVs− and EVr−EVs+ types (Supplementary Fig. 6). Thus, divisively normalized value signals in mOFC were somewhat related to the risk attitude of monkeys.

## Discussion

Normalization is a canonical computational process widely observed in the domain of sensory processing[35–38], from early sensory representation to higher-order phenomena such as multisensory integration[38]. Here, we found that mOFC neurons employ divisively normalized value coding during an economic decision-making task. This is the first demonstration of the common normalization computation in frontal decision circuits. This normalization depended on task context: the response sensitivity of mOFC neurons to reward values was stronger when animals made choices in a free choice task. The normalization model outperformed other models in the free choice task (Fig. 4b) and performed equally well in the forced choice task compared to other models (Supplementary Fig. 5a); however, the normalization model better explained relative value coding in free versus forced choice contexts (Fig. 7 and Supplementary Fig. 4). These results suggest that the mOFC is critical for economic decision making when comparing alternative rewarding options.

The orbital and ventromedial part of the frontal lobe is composed of a large set of heterogeneous cortical regions. The orbital network receives sensory inputs from several modalities, presumably to relate them to item preferences or values. The medial network is widely believed to provide the major cortical output related to emotion and mood[4]. mOFC seems to be functionally situated in an intermediate position between these two network areas. For example, the activity of vmPFC neurons is thought to combine information about option values[25] and satiety level[26,27], and might well be related to stochastic preferences and choice behavior[27]. Perhaps the lOFC is the brain locus that signals the relative values of items animals recently encountered[11,12]. Other evidence suggests that the frontal pole region may signal an animal's decisions specifically after monkeys choose in "free choice" trials at the time of outcome delivery[39]. Overall, our present results suggest that mOFC neurons represent information that combines aspects of both medial and orbital function to yield normalized value signals, but primarily during so-called free choice trials.

In terms of relative value coding, the signals observed in mOFC (Fig. 2) are similar in principle to those in lOFC[11,12,16,40]. They represent relative value signals among the set of possible outcomes in a block of trials. Although it is not known what type of normalization is employed in lOFC neurons, one possible distinction between mOFC and lOFC in terms of relative value coding is the dependency on the behavioral context. The free choice-specific relative value coding in mOFC (Fig. 7) may represent a key difference from lOFC neurons, where relative value signals are observed even in non-choice situations during classical conditioning[12]. The enhancement of value signals during free choices is consistent with the finding in human mOFC that value signals are specifically observed when subjects evaluate economic options[41]. Enhanced value signals during free choice have also been found in the activity of monkey amygdala neurons[42,43], which is connected to the orbitofrontal cortex. Thus, mOFC could regulate behavioral sensitivity to reward values[44] (i.e., gain) depending on behavioral context.

Many normative models of choice assume that values are represented in an absolute manner[8–10]. Under absolute value coding, the neuronal discharge rate does not depend on what other values might have been encountered. In contrast, under normalized coding, the neuronal discharge rates reflecting a given value will depend on factors such as other present and past values. Relative value signals have been examined in single neuron activity in regions including prefrontal[11,12,16] and parietal cortex[34], and striatum[45] with relatively few examples of studies using human neuroimaging[14,46]. This discrepancy in the literature may arise from multiple differences in species and methodologies. Blood oxygenation level dependent (BOLD) activity is often examined using a linear regression approach which would be unable to identify nonlinear normalized signals, but instead would tend to identify such signals as mixtures or positive and negative regression coefficients[47,48]. Indeed, there have been only a couple efforts to search specifically for nonlinear normalization-type representations in the BOLD signal[49] and these efforts have been successful to some degree.

How are divisively normalized value signals related to the monkey's choice behavior? This still remains an open question, but one possible explanation is that divisive normalization, which yields decreased neural value sensitivity with increases in total values, would yield decreased sensitivity to increase in values in behavior, known as the subjective value or utility. Recent works in economic decision-making studies hypothesized that neuronal activity is linearly correlated with subjective value functions, an approach successfully examined in human imaging[21,50] and monkey electrophysiology[51,52] experiments. Our results suggest that the divisively normalized value signals in mOFC were at least related to the risk attitudes observed in corresponding monkey behavior. However, the precise relationship between normalized value coding in mOFC and behaviorally derived subjective values remain unknown, and further experimental and theoretical work will be required to link behavioral and neural observations for relative value coding.

The efficient coding hypothesis assumes that the neural code adapts efficiently to the present behavioral context, and that neurons change their firing rates in order to utilize their entire dynamic range during encoding[13]. Efficient coding requires input–output functions to use the entire response range to represent the stimulus distribution[53]. In the domain of sensory systems for perception, a large literature supports the hypothesis that normalization is employed to achieve efficient coding[17]. Moreover, a recent finding by Coen-Cagli et al.[54] shows that normalization processes in primary visual cortex can be flexibly gated depending on the sensory context. In contrast to the sensory domain, only a couple of direct and indirect tests have been conducted to examine the implementation of efficient coding in decision making[11,12,14,34]. Our current study highlights that

value-based divisive normalization occurs in frontal decision circuits; furthermore, the modulation of this normalization by the behavioral choice context suggests that the flexible gating of contextual information occurs in both sensory and decision-related computations. The existence of such context-specific value normalization suggests that the mOFC contributes to the construction of value critical for economic decision making.

## Methods

**Subjects and experimental procedures**. Two rhesus monkeys were used (DE, 7.5 kg, 6 years; HU, 8.0 kg, 6 years). All experimental procedures were approved by the New York University Institutional Animal Care and Use Committee and performed in compliance with the US Public Health Service's Guide for the Care and Use of Laboratory Animals. Each animal was implanted with a head-restraint prosthesis and a scleral eye coil[55]. Eye movements were measured using a scleral coil at 500 Hz. Visual stimuli were generated by cathode ray tube (CRT) 30 cm away from the monkey's face when they were seated. After subjects practiced the lottery task for 6 months, they were proficient at making choices of risky and safe options[30].

**Electrophysiological recording**. We used conventional techniques for recording single neuron activity from mOFC. Monkeys were implanted with recording chambers (Crist Instrument) targeting the medial part of the prefrontal cortex, centered midline and 30 mm anterior in stereotaxic coordinates. Chamber location was verified using anatomical magnetic resonance imaging (Siemens). In each recording session, a stainless steel guide tube was placed within a 1 mm spacing grid (Crist Instrument), and a tungsten microelectrode (1–2 MΩ, FHC) was passed through the guide tube. The electrode was lowered until reaching close to the bottom of the brain after passing through the cingulate cortex. Electrophysiological signals were amplified, band pass filtered and monitored and single neuron activity was isolated based on spike waveform. We recorded 182 mOFC neurons from four hemispheres of two monkeys (Supplementary Fig. 2). All single neuron activity was sampled when the activity of an isolated neuron showed a good signal-to-noise ratio (>3). No blinding was made. Sample size to detect the effect size (number of the recorded neurons, number of the recorded trials in a single neuron and number of the monkey used) was in estimated reference to the previous study[34].

**Cued-Lottery task**. Animals performed one of two visually cued saccadic choice tasks: forced choice and free choice trials. The color of the central target indicated forced choice (red or yellow, indicating which of the two options was rewarded) or free choice (gray) trials.

Forced choice trials: If the central fixation target was red or yellow, monkeys were required to choose the color-matched target in order to receive any reward. Each trial started with a 0.3 s 500 Hz tone, after which the monkey had 1.0 s to align gaze to within 2° of a 1° diam central fixation target. After fixating for 0.4 s, two peripheral 8° pie charts providing information about reward magnitude for each of the two options were presented for 0.5 s, 8° to the left and right of fixation. Red and yellow 1° choice targets appeared at these same locations 0.1 s after cue offset. At 0.3 s later, the fixation point disappeared, cueing saccade initiation. A correct saccade that shifted gaze to within 3.5° of the choice target matching the color of the fixation target could yield a water reward. Red and yellow colors were randomly assigned to fixation and peripheral targets on each trial. When the central fixation cued a "safe" reward, animals received the reward indicated by the pie chart if they shifted gaze to the associated target. When the fixation color cued a choice to the risky target, animals received the reward indicated by the pie chart with a probability of 0.5, otherwise no reward. A 1 and 0.1 kHz 0.3 s tone indicated reward and no-reward outcomes, respectively. A high tone preceded a reward by 0.2 s. A low tone indicated that no reward would be delivered, but that the task had been performed correctly. If animals chose a non-match target, the trial was aborted. A 2.0 s inter-trial interval followed. Aborted and error trials were presented again.

Free choice trials: Trials began with the onset of a gray central fixation target. As in the forced choice trials, pie charts indicated the magnitude of safe and risky rewards. After offset of the fixation target, animals were free to choose by shifting gaze to either target. The locations of the risky and safe targets were fixed during a block of trials.

**Pay-off and block structure**. Pie-charts indicated reward magnitudes from 60 to 600 μl in 60 μl increments (Fig. 1a). A 5 μl reward was signaled by a blank pie chart. During data collection, blocks of 86 trials were presented, in random order, built from one of the 4 payoff blocks (Fig. 1c). The first 36 trials (6 repeats times 6 conditions, five risky and one safe choices) in a block were forced choice trials. Then, 50 free choice trials (10 of each 5 type) followed. During a block the safe option was fixed and the magnitude of the risky option varied randomly across its 5 possible values (Fig. 1b). The middle-valued risky target always offered a reward of the same expected value as the safe target in that block. A new block with a new payoff was then presented.

**Calibration of reward supply system**. The precise amount of liquid reward was controlled and delivered to the monkeys by the use of solenoid valves. A 18-gauge

tube (0.9 mm inner diameter) was attached to the tip of the delivery tube to reduce the trial-by-trial variability of reward supply. The amount of reward in each payoff block was calibrated by measuring the weight of water to 0.002 g precision (hence 2 µl) in single trial basis. Note that if we used bigger diameter tubes attached to the tip (4 mm inner diameter), the variability of reward sizes increased dramatically.

**Statistical analysis.** For statistical analysis, we used the statistical software package R (http://www.r-project.org/). All the statistical tests we used were two tailed.

**Behavioral analysis.** We examined whether monkey's choice behavior depended on the relative value of risky and safe options. In each payoff block, five risky options were paired with one safe option as five types of lottery pairs (LP) in terms of their relative values; expected values of risky options were either considerably larger than the safe option (LP5), slightly larger (LP4), equal (LP3), slightly smaller (LP2) or considerably smaller (LP1). We examined whether the percentage of risky choices in each payoff block changed in parallel with the relative values of risky option against safe options by plotting the percentage of risky choices in each of the four PBs (Fig. 1d). The behavioral results have been previously reported in Yamada et al.[30] In addition, we quantified the percentages of correct trials (i.e., non-aborted trials) in each of the 20 lottery pairs and saccadic reaction times (latency of responses after the fixation point disappeared).

**Neuronal analysis.** We analyzed neuronal activity during three task periods: cue period (1.0 s window after cue onset), saccade period (1.0 s window after saccade onset) and feedback period (1.0 s after feedback onset). The maximum firing rate of a neuron was defined as the maximal firing rate in a trial during the three task periods. The baseline firing rate of a neuron was defined as the average firing rate in the 600 ms window just before cue onset. To display peri-stimulus time histograms of neural activity (Fig. 2a, c), the average activity curves were smoothed using a 100 ms Gaussian kernel ($\sigma = 100$ ms).

**Relative value signals.** To prescreen relative value signals in the activity of mOFC neurons without normalization equations, we first determined whether mOFC neurons signal relative value by using a variable selection approach. Neuronal discharge rates ($F$) were fitted by a linear combination of the following variables:

$$F = b_0 + b_1 \mathrm{EVr} + b_2 \mathrm{EVs} + b_3 \mathrm{Fb} \tag{1}$$

where EVr and EVs were the expected values of risky and safe options, respectively. The Fb, feedback type, took scalar values (1, 0) in reward and no-reward trials and was included only during the feedback period. $b_0$ was the intercept. Among many possible combinations of these variables ($b_0$, EVr, EVs, Fb), we selected one model that contained the combination of variables showing minimal AIC:

$$\mathrm{AIC(Model)} = -2\log(L) + 2k \tag{2}$$

where $L$ is the maximum likelihood of the model and $k$ is the number of free parameters in the model. If the selected model contained EVr and EVs and their coefficients showed opposite signs (i.e., positive $b_1$ and negative $b_2$ or negative $b_1$ and positive $b_2$), the discharge rates were regarded as being modulated by the relative value of risky and safe options. Two types of relative value modulation (positive $b_1$ and negative $b_2$: EVr+EVs−, or negative $b_1$ and positive $b_2$: EVr−EVs+) were identified. Neuronal activity during free choice trials was used for this classification.

**Choice signals.** To examine whether the mOFC neurons signal the spatial choice of monkeys, we also analyzed neuronal discharge rates by using a variable selection. The model used for this approach included an additional parameter for spatial choice location:

$$F = b_0 + b_1 \mathrm{EVr} + b_2 \mathrm{EVs} + b_3 \mathrm{Fb} + b_4 \mathrm{Cho} \tag{3}$$

where Cho took scalar values (1, 0) in the trials if the monkey chose the left and right targets, respectively. Fb was included only during the feedback period. Among all possible combinations of these variables, we selected one model that contained one combination of variables showing minimal AIC. If the selected model had $b_4$ without $b_1$–$b_3$, the discharge rates were regarded as being exclusively modulated by the left–right target choice. Note that the percentage of the activity modulated by the relative values of options was not different than that estimated using Eq. 1.

**Normalization models.** 1. Advanced fractional model: The normalization equation was originally proposed to describe nonlinear response properties in early visual cortex, and later discovered to characterize neural activity in other sensory processing areas and modalities[17]; recent work showed that normalization extends to reward coding in parietal cortex[13]. Under the condition where a subject chooses one option from two alternatives, the neuronal response to option 1, $R_1$, depends on the expected value of the two options:

$$R_1 = R_{\max} \frac{\beta + \mathrm{EV}_1}{\sigma + \mathrm{EV}_1 + \mathrm{EV}_2} \tag{4}$$

where $\mathrm{EV}_1$ and $\mathrm{EV}_2$ are the expected values of option 1 and 2, respectively. $R_{\max}$, $\beta$ and $\sigma$ are free parameters. $R_{\max}$ determines the maximal level of neural activity. $\beta$ and $\sigma$ determine the relative contribution of the expected values to neuronal response, with $\beta$ governing the theoretical level of activity when no cue stimulus appeared and $\sigma$ determining the sensitivity of neuronal responses to the expected values (large $\sigma$ means low sensitivity).

In the lottery task, the two options were defined as risky and safe options, respectively, as follows. If the activity of the relative value coding neuron showed positive and negative regression coefficients to the expected values of the risky (EVr) and safe (EVs) options, respectively (i.e., EVr+EVs− type), $\mathrm{EV}_1$ and $\mathrm{EV}_2$ were the EVr and EVs, respectively. If the neuronal activity showed negative and positive regression coefficients to EVr and EVs, respectively (i.e., EVr−EVs+ type), $\mathrm{EV}_1$ and $\mathrm{EV}_2$ were the EVs and EVr, respectively.

2. Simple fractional model: The simple fractional model is a simplified form of the normalization equation presented above. In the model, neuronal response to option 1, $R_1$, is given by:

$$R_1 = R_{\max} \frac{\mathrm{EV}_1}{\mathrm{EV}_1 + \mathrm{EV}_2} + b \tag{5}$$

As above, $\mathrm{EV}_1$ and $\mathrm{EV}_2$ are the expected values of options 1 and 2, respectively. $R_{\max}$ determines the maximal level of neural activity and $b$ is the baseline firing rate when no cue stimulus appears. $R_{\max}$ and $b$ are free parameters. In the lottery task, if the activity of the relative value coding neuron showed positive and negative regression coefficients for EVr and EVs, respectively (i.e., EVr+EVs− type), $\mathrm{EV}_1$ and $\mathrm{EV}_2$ were EVr and EVs, respectively. If the neuronal activity showed negative and positive regression coefficient to EVr and EVs, respectively (i.e., EVr−EVs+ type), $\mathrm{EV}_1$ and $\mathrm{EV}_2$ were the EVs and EVr, respectively.

3. Difference model: In the difference model, neuronal response, $R_1$, is a simple linear function of the value difference between the two options:

$$R_1 = G(\mathrm{EV}_1 - \mathrm{EV}_2) + b \tag{6}$$

$G$ determines the magnitude of neural response to value difference (i.e., gain), and $b$ is the baseline firing rate when the expected values of options are equal or no cue stimulus appeared. $G$ and $b$ were free parameters. This model is often used in reinforcement learning models[33] and race-to-threshold models[32].

4. Range normalization model: A phenomena called range adaptation has been observed in the activity of lateral OFC neurons[11,12]. The normalization equation has not been clearly established to describe this type of neuronal activity, but we assume the following equation as a range normalization model; this formulation has been found to describe the activity modulation in lOFC neurons observed previously (Fig. 3, right panel). In range adaptation, the relative value of an option depends on the range of reward values of all options available in a block of trials. In the model, neuronal response to option 1, $R_1$, depends on the relative position in the distribution of values:

$$R_1 = R_{\max} \frac{\mathrm{EV}_1 - V_{\min}}{V_{\max} - V_{\min}} + b \tag{7}$$

Where $\mathrm{EV}_1$ was the expected value of option 1. $V_{\max}$ and $V_{\min}$ are the largest and smallest reward values in a block of trials, respectively. The denominator defines the range of the reward values in a block of trials, while the numerator indicates relative position of the expected values of option 1 according to the minimal value in the distribution of values, and thus, they represents the relative position of the expected values of option 1 as a percentage in the distribution of values in a block of trials. $R_{\max}$ determines the semi-saturating firing rate and $b$ is the baseline firing rate when no cue stimulus appears. $R_{\max}$ and $b$ are free parameters. Note that in this assumed model, the value is not normalized by the values of other options, but rather by the range of reward values available in a block of trials.

In the lottery task, $V_{\max}$ and $V_{\min}$ in first payoff block were 240 and 0 µl (no reward), respectively, and hence the value range was 240 µl. In the second payoff block they were 360 and 0 µl, and the value range was 360 µl. In the third and fourth payoff blocks, value ranges were 480 and 600 µl, respectively. As seen in Fig. 3 ("Range normalization model"), this model formulation predicts a block-dependent range adaptation in neural firing rate: the predicted sensitivity of neuronal firing rate to risky values decreases as value range increases according to payoff block. Indeed, the output of the model was very similar to previously published results (Figs. 5B and 6B in Padoa-Schioppa, 2009)[11].

**Other possible alternative models.** 5. Expected values of risky options: In the model, neuronal response, $R_1$, is a simple linear function of the expected values of risky options:

$$R_1 = a\mathrm{EVr} + b \tag{8}$$

$a$ determines the magnitude of neural response to the expected values of risky options and $b$ is the baseline firing rate. $a$ and $b$ are free parameters.

6. Expected values of safe options: In the model, neuronal response, $R_1$, is a simple linear function of the expected values of safe options:

$$R_1 = a\,\mathrm{EVs} + b \qquad (9)$$

$a$ determines the magnitude of neural response to the expected values of safe options and $b$ is the baseline firing rate. $a$ and $b$ are free parameters.

7. Expected values of chosen options: In the model, neuronal response, $R_1$, is a simple linear function of the expected values of options monkeys chosen in the current trials:

$$R_1 = a\,\mathrm{EVchosen} + b \qquad (10)$$

$a$ determines the magnitude of neural response to the expected values of chosen options (EV chosen) and $b$ is the baseline firing rate. $a$ and $b$ are free parameters.

8. Choice of risky options: In the model, neuronal response, $R_1$, is a simple function of whether monkeys chose risky option or not in the current trials (RiskyCho):

$$R_1 = a\,\mathrm{RiskyCho} + b \qquad (11)$$

Where RiskyCho took scalar values (1, 0) in the trials if monkey chose risky and safe options, respectively; $a$ determines the magnitude of neural response to the choice of risky option and $b$ is the baseline firing rate. $a$ and $b$ are free parameters.

9. Null model: In the model, neuronal response, $R_1$, is only a function of the mean firing rate:

$$R_1 = b \qquad (12)$$

$b$ determines the mean firing rate. $b$ is a free parameter.

10. An artificial model: In the model, neuronal response, $R_1$, is a function of the expected values of risky options in each payoff block:

$$R_1 = a_1\,\mathrm{EVr} + b_1 + a_2\,\mathrm{EVr} + b_2 + a_3\,\mathrm{EVr} + b_3 + a_4\,\mathrm{EVr} + b_4 \qquad (13)$$

$a_1$–$a_4$ determine the magnitude of neural response to the expected values of risky options in the payoff block number 1 to 4, respectively. $b_1$-$b_4$ are the baseline firing rate in the payoff block number 1 to 4, respectively. $a_1$-$a_4$ and $b_1$-$b_4$ are free parameters.

To evaluate the relationship between our primary relative value models and other known characteristics of OFC value representations, we calculated correlation coefficients between the relative expected values (derived from the fractional model, difference model and range model) and other possible known explanatory variables[51], such as the expected values of risky options, expected values of safe options, expected values of chosen options and choice of risky options (Supplementary Table 1). Note that these relative expected values were defined with no-free parameters, since the estimated free parameters mentioned above were different neuron by neuron.

**Fitting and selection of normalization models**. To identify the best structural model to describe the activity of mOFC neurons, we examined the four relative expected value models as well as six other alternative models. We fitted the 10 alternative models to the activity of each single neuron that demonstrated relative value coding as defined by our regression analyses. In each of the models, we estimated a combination of the best-fit parameters to explain neuronal discharge rates by using the statistical software package R. Best-fit parameters were estimated in each epoch of the activity of the neuron based on single trial firing rates. We used the nls() function with random initial values (repeated 100 times). In this function, a set of parameters that minimize non-linear least squared values were estimated. Across the population, the best-fit model showing minimal AIC was selected by comparing AIC differences among models. If the AIC differences against the nine other models was significantly different from zero at $P < 0.05$ by one-sample $t$-test, the model was defined as the best model. The estimated parameters in the best-fit model were compared by using parametric and non-parametric tests, respectively, with a statistical significance at $P < 0.05$. Note that models were separately fitted to the free choice and forced choice trial data.

**Evaluation of model performance**. To evaluate model performance, we estimated the percentages of variance explained, which is defined as one minus percentage of the residual variances out of total variances. The percent variance explained in each neuron was estimated based on either single trial data or mean responses data (segregated by the 20 lottery pair conditions). The mean response-based percent variance explained is similar in principle to explainable variance[56]. To validate the accuracy of estimation and model selection, we performed two-fold cross-validation (i.e., half split) in each of the model fits as follows. First, we prepared training data and test data, by randomly dividing the data in half in each of 20 lottery pairs. Models were fitted to the training data and best-fit parameters were estimated. By using these estimated parameters, percentages of variance explained were estimated for the test data.

**Model fit during forced choice trials**. During the forced choice trials, monkeys were required to choose the color-matched targets. If they selected the other target, the trial was aborted and no reward was received. There are two alternative ways in

which mOFC neurons could encode the expected values of the risky and safe options in such a situation and we tested both of them.

One possibility is that in the forced choice trials, mOFC neurons encode the expected values of both risky and safe option in the same manner as in the free choice trials (assumption 1). To examine this possibility, we fitted all four models using the same assumptions as in free choice trials. The other possibility is that mOFC neurons only encode the expected value of options that are available to the chooser. In this case, they would encode the value of the color-matched target, but they would encode the value of non-selectable option as 0 independent of the reward size cued for non-matched target (assumption 2). This is because no matter what reward size is associated with the non-selectable cue, choosing it gives no reward since a trial is aborted after the choice of a non-matched target. To examine this possibility, we fitted the models with a slight modification—the value of non-selectable options was set to zero. For the option forced to choose, those values were defined as those in the free choice trials (i.e., the values cued by pie chart).

We fitted the four alternative relative value coding models to the data under both of these two assumptions and compared AIC values (Supplementary Fig. 5). Percentage of the variance explained by the models was compared using the paired $t$-test with a statistical significance at $P < 0.05$ (Supplementary Fig. 4).

**Model fit including behavioral measures**. To examine whether the activity difference between free and forced choice trials could be explained by differences in state (i.e., motivation or attention) rather than differences in context, we fitted the following three modified versions of advanced fractional models. The models were simultaneously fitted to both free and forced choice trial data.

$$11. \quad R_1 = R_{\max}\frac{\beta + \mathrm{EV}_1}{\sigma + \mathrm{EV}_1 + \mathrm{EV}_2} + a\,\mathrm{Context} \qquad (14)$$

$$12. \quad R_1 = R_{\max}\frac{\beta + \mathrm{EV}_1}{\sigma + \mathrm{EV}_1 + \mathrm{EV}_2} + a\,\mathrm{Percent\ correct} \qquad (15)$$

$$13. \quad R_1 = R_{\max}\frac{\beta + \mathrm{EV}_1}{\sigma + \mathrm{EV}_1 + \mathrm{EV}_2} + a\,\mathrm{RT} \qquad (16)$$

Where Context took scalar values (1, 0) in the free and forced choice trials, respectively. Percent correct was the percentages of the correct trials estimated in each of 20 lottery pairs in a given neuronal recording period. RT was the saccadic reaction time after the central fixation target disappeared. $R_{\max}$, $\beta$, $\sigma$ and $a$ are free parameters. We compared AIC to define which model best explained the activity difference between free and forced choice trials.

**Data availability**. All relevant data are available from the authors.

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

## Acknowledgements

The authors wish to express appreciation to Masayuki Matsumoto. This research was supported by the National Institutes of Health grant number DA038063 (P.G.) and by JSPS KAKENHI Grant Number JP:15H05374 (H.Y.) and Takeda Science Foundation (H.Y.).

## Author contributions

H.Y., K.L. and P.W.G. designed the research. H.Y. conducted experiments. H.Y. analyzed data. H.Y., K.L., A.T. and P.W.G. evaluated the analyzed results. H.Y. wrote the manuscript. K.L. wrote a part of the manuscript. H.Y., K.L., A.T. and P.W.G. edited and approved the final manuscript.

## Additional information

**Competing interests:** The authors declare no competing financial interests.

