## [Peer Review File · Nature Communications]

Reviewers' comments:

Reviewer #1 (Remarks to the Author):

Yamada and colleagues tested whether neuronal value-coding responses in medial orbitofrontal cortex (mOFC) during economic choice are consistent with a canonical divisive normalization computation. Monkeys made saccade choices between risky and safe options under a schedule that systematically varied relative expected value. Activity in 28 of 101 neurons regressed significantly on relative expected value. Model fitting showed that a divisive normalization model, compared to plausible alternatives, best described these responses. Importantly, this value normalization was significantly reduced during forced choice. The implications are that mOFC neurons may be important in the computation of relative value during economic choice, and that a divisive normalization mechanism known from sensory systems operates in prefrontal valuation systems in a task-dependent manner.

This is a well-conducted study that will likely influence our thinking about valuation processes in orbitofrontal cortex. It also has wider implications for understanding both cortical computation and economic choice mechanisms. Accordingly, the data are of interest to specialists and broader audiences. In my view, the main strengths of the paper are (i) the first quantitative demonstration of divisive normalization in OFC; (ii) neuronal comparison between free choice and forced choices; (iii) exploration of poorly understood medial OFC. A slight drawback is the relatively lower number of studied neurons, compared to previous studies on OFC; however, I think this should not detract from the merits of the paper. I do have several concerns though, that I feel have to be addressed convincingly before publication. All concerns can be addressed by careful analysis and changes to text.

Major concerns

1. It is usually accepted that OFC neurons encode subjective value rather than objective value, and this has been demonstrated for risky rewards (Raghuraman and Padoa-Schioppa, J Neurosci, 2014). In their previous paper on behavioral data from the present experiment (Yamada et al., PNAS, 2013), the authors demonstrated that animals' risk attitudes influenced choice: animals were "slightly, but significantly, risk averse". Therefore, I was surprised that in the present manuscript the authors did not explicitly model subjective values as a regressor that incorporated animal-specific risk attitudes but instead relied exclusively on objective expected values. Do mOFC neurons encode objective or subjective value? Do the paper's main results on divisive normalization still hold with subjective values that incorporate risk attitudes? To answer these critical questions conclusively, it may be necessary to analyze specific trial types where expected values of safe and risky options were identical, and where objective and subjective value coding can be distinguished.
2. I was also surprised that canonical signals observed in (more lateral) OFC during choices were not explicitly tested, including offer value, chosen value and risky choice (Raghuraman and Padoa-Schioppa, J Neurosci, 2014). For comparison with existing frameworks on OFC coding, it is important to know (i) how strongly the presently reported relative expected value regressor was correlated with the alternative explanatory variables offer value of risky or safe option, chosen value, and risky choice (the latter being a dummy variable indicating whether or not the risky option was chosen); and (ii) whether relative expected value indeed provides a better account of mOFC responses than these alternatives. I suggest performing supplementary regression analyses and goodness-of-fit comparisons to address these issues.
3. To fully interpret differences in neuronal coding between free and forced tasks, we require more detailed description of the animals' behavior. For example, were there differences in performance

(percentage of correct trials) and reaction times that might indicate different levels of motivation between tasks? If such differences exist, could they partly explain the diminished divisive normalization in forced trials? In particular, to test reward expectation and related motivation, did reaction times correlate with upcoming reward magnitudes in both tasks, and if so were these correlations similar between tasks? Similar correlation could strengthen the authors' conclusion that weaker divisive normalization in forced trials is related to the absence of choice requirement (rather than other factors such as motivation).

4. The hypothesis that guides the present study is that divisive normalization is a canonical computation implemented by cortical neurons. Accordingly, the authors test the extent to which mOFC responses can be described by the formal divisive normalization model by fitting the relevant parameters to individual neuron's activities. This approach contrasts with other approaches whereby model parameters for reward value functions are derived from fits to observed choices, and then these fitted reward values are used as regressors for neuronal activity (e.g. some of the temporal discounting and reinforcement learning studies cited in the manuscript). This latter approach has the advantage that neuronal value signals are demonstrably related to the observed behavior, as the model parameters were fitted to behavior. I wonder whether the authors can find a way to relate the presently observed divisively normalized expected value signals to the monkeys' behavioral choices. At least it would be helpful to discuss the behavioral relevance of these signals and explain the advantages conferred by divisively normalized values for adaptive decision-making.

Minor concerns

5. From the description it seems that the ratio of safe to risky choices in the forced task was not matched to the animals' specific preferences during the free task. Could this difference have affected the neuronal coding?

6. I do not understand what was tested in lines 208-212 to rule out an adaptation process. Please provide a more detailed description.

7. So far, few studies have directly compared neuronal coding of valuation and decision variables in free choice and forced tasks. The demonstration of diminished relative valuation in forced trials is a clear strength of the present study. A few recent papers on amygdala neurons in economic choice reported similar phenomena, including relative specificity of value signals to free choice (Hernadi et al., *Nature Neuro*, 2015) and diminished neuronal adaptation during forced trials (Grabenhorst et al., *eLife*, 2016), which may inform the discussion in lines 273-283.

8. It would be helpful to report the amount of explained variance by the value regressor for the example neuron.

9. The plot in Fig. 6b showing transition in neuronal coding from forced to free trials is particularly striking, as it clearly indicates the rapidity of change in mOFC value coding. Could the authors quantify this difference statistically, e.g. demonstrating significant change in coefficients from last forced to first free trial block?

Reviewer #2 (Remarks to the Author):

Yamada H et al., Free choice shapes normalized value signals in medial orbitofrontal cortex

The authors analyze data recorded in mOFC in a task where monkeys chose between sure and probabilistic outcomes. They found that animals chose the probabilistic outcome as its expected value increased. When they examined neural activity they found that it scaled relative to the other offers available in a block of trials. They fit several models to the neural activity and found that the advanced fractional model, a divisive normalization model, fit the data best at the population level. In forced choice trials this was also true, although the scaling was less pronounced.

The coding strategies used in areas that represent choice values have been explored in much less detail than the coding strategies used in primary sensory areas. This paper begins to make progress on this question by identifying aspects of the coding in mOFC. The paper is clearly written and straightforward. A few details were unclear or I missed them.

1. What was the likelihood model? Gaussian? The AIC differences seemed relatively small. Were these based on single trials, or averages across trials for each condition? Also, in Fig. 4b, is this across task epochs? Or does each neuron contribute just once to this? How frequently were neurons significant in more than one epoch? Were the 81 neurons that were analyzed neurons that were significant in all epochs, or at least one?
2. Does the divisive normalization model perform better than a model which floats the intercept and slope for each block?
3. Is the neuron in Fig. 4 the best typical example?

Reviewer #3 (Remarks to the Author):

The manuscript reports on new experiments and analysis, addressing the role of medial Orbitofrontal Cortex (mOFC) neurons in coding value signals. The study is a nice follow up on previous work by the same lab on 'relative' value coding and divisive normalization, and reports two main findings: 1) mOFC neurons can be described by divisive normalization, in which the expected value of a choice option is normalized by the expected value of an alternative option (defined as free-choice context); 2) the normalization seems abolished when the alternative option has zero value (defined as forced-choice context).

Overall the manuscript is interesting and well written. The problem is clearly stated and placed in the context of relevant literature; the approach is well justified; and the findings are original to the best of my knowledge, and would be of interest to others in the neuroscience community. However, my main concern is that the results are not convincing and, in my opinion, do not support quantitatively the proposed conclusion, as detailed below. I have also a couple of minor comments on the discussion of related literature.

Major

To support the main finding of context dependent normalization in mOFC, the authors propose a model of single-neuron firing rate based on divisive normalization (equation on line 147), fit it to data and compare it to alternative models. I find a number of issues with the analyses and the results

reported.

1) Poor fits.

Primarily, the model fits are poor even for the example cell (Figure 4), and a disaster across the population (in Supplementary Figure 3, the variance explained is <10%!). The authors compare models using relative AIC scores (more on this below), but they do not report quantities that would more clearly capture fit quality. One obvious choice is R-square or variance explained, which is relegated to just one Supplementary figure where it shows that the model only captures <10% of the variance of the data. This may be in line with other work in OFC (of which I am no expert) and if so it would be useful to state it; but, even then, I find it hard to build a case that these neurons encode relative value in a context-dependent manner when relying on a model that misses more than 90% of the response.

I would suggest the authors dig into the qualitative aspects of neural responses that the are missed by the model, and then explore variants of divisive normalization that could improve model fits. For instance, the model seems to miss the strong saturation at moderate and high EVr, and perhaps adding an exponent to the denominator could correct that.

Also a couple of technical suggestions to improve fit quality: I would normalize the rates by their across-trial variances (eg see Cavanaugh et al 2001 J Neurophys), or consider some other noise model that is more accurate for cortical neural activity – if I understand correctly the Methods, the current fits assume additive noise of constant amplitude (not stated explicitly, but seems implied). Second, rather than percent explained variance, the authors should report percent ‘explainable’ variance, ie. normalized to a noise ceiling imposed by the limited number of measurements. Related: for completeness, I would suggest reporting the number of repeats per condition that are used to measure firing rates; this will become relevant when computing the noise ceiling.

2) Imperfect model comparison.

First, I suggest the authors use a different metric to compare models. Intuitively, AIC penalizes any parameter by the same amount, which is fine for models that are linear in the parameters (and under the assumption of Gaussian observation noise). But the normalization models are not linear in the parameters, which means that, for instance, the additional model expressivity (or complexity) offered by σ in model 1 is not the same as ‘b’ in the other models. The most straightforward choice would be cross-validated percent explainable variance (ie normalized model loglikelihood). Another, more principled option, is Bayesian Model Comparison, or the ‘widely applicable information criterion’ (WAIC), as they take into account the full posterior over parameters. The choice will depend on the specifics of the data (primarily, number of data points available).

Second, as the fit quality is so poor, one is left wondering if a ‘null’ model that assumes constant response (ie just 1 free parameter) would compare better than all of the proposed models, once accounting properly for model complexity.

Minor

1) I found the statement in the first Discussion section (starting on line 258) a bit too strong. The authors suggest that “the clear distinction between mOFC and IOFC in terms of relative value coding is the dependency on the behavioral context.” In this manuscript, the authors use an experimental design that probes different contexts explicitly, and quantitative analysis based on model comparison, to conclude that there is context dependence. As far as I understand the cited reference on IOFC, in

that experiment there is only one context, so it is hard to conclude so strongly that IOFC would not show context dependence, if one were to test it.

2) Overall, I found the discussion of related work on efficient coding and normalization in sensory processing, light and rushed (please check for typos). In particular, the finding that divisive normalization operates under one context but is abolished under a different context has been shown recently in visual cortex (Coen-Cagli et al 2015 Nature Neuro) and described by a model equivalent to the one dubbed here "assumption 2" where the context term in the normalization signal is set to zero.

3) An interesting prediction of the advanced fractional model is that the baseline firing rate of a neuron of the class EVr+ EVs- should be modulated by the safe option value even in the absence of a risky option. Would it be possible to test this in the existing data?

4) For clarity, it would be useful to state explicitly what is the time window used to estimate the firing rates that went into the models (eg around line 161). Also, on line 205, I would remind the reader what 'regression coefficients' refer to.

Typos: line 80, 'situation*al*'; line 293, 'studies' repeated twice

Point-by-point reply to the reviewer's comments

We greatly appreciate the abundant insightful comments from the three reviewers. We have carefully reviewed the comments, and have fully revised our manuscript after performing the additional analyses the reviewers suggested. We believe that addressing the comments and advice from the three reviewers have made our results clearer and more convincing, thereby strengthening our manuscript significantly. Below we provided our point-by-point responses to their comments.

Subject: Manuscript NCOMMS-17-05356

"Free Choice Shapes Normalized Value Signals in Medial Orbitofrontal Cortex"

Reviewers' comments:

Reviewer #1 (Remarks to the Author):

Yamada and colleagues tested whether neuronal value-coding responses in medial orbitofrontal cortex (mOFC) during economic choice are consistent with a canonical divisive normalization computation. Monkeys made saccade choices between risky and safe options under a schedule that systematically varied relative expected value. Activity in 28 of 101 neurons regressed significantly on relative expected value. Model fitting showed that a divisive normalization model, compared to plausible alternatives, best described these responses. Importantly, this value normalization was significantly reduced during forced choice. The implications are that mOFC neurons may be important in the computation of relative value during economic choice, and that a divisive normalization mechanism known from sensory systems operates in prefrontal valuation systems in a task-dependent manner.

This is a well-conducted study that will likely influence our thinking about valuation processes in orbitofrontal cortex. It also has wider implications for understanding both cortical computation and economic choice mechanisms. Accordingly, the data are of interest to specialists and broader audiences. In my view, the main strengths of the paper are (i) the first quantitative demonstration of divisive normalization in OFC; (ii) neuronal comparison between free choice and forced choices; (iii) exploration of poorly understood medial OFC. A slight

drawback is the relatively lower number of studied neurons, compared to previous studies on OFC; however, I think this should not detract from the merits of the paper. I do have several concerns though, that I feel have to be addressed convincingly before publication. All concerns can be addressed by careful analysis and changes to text.

Response: Thank you for such a positive summary of our work. We carefully address the specific concerns of the Reviewer below.

Major concerns

1. It is usually accepted that OFC neurons encode subjective value rather than objective value, and this has been demonstrated for risky rewards (Raghuraman and Padoa-Schioppa, J Neurosci, 2014). In their previous paper on behavioral data from the present experiment (Yamada et al., PNAS, 2013), the authors demonstrated that animals' risk attitudes influenced choice: animals were "slightly, but significantly, risk averse". Therefore, I was surprised that in the present manuscript the authors did not explicitly model subjective values as a regressor that incorporated animal-specific risk attitudes but instead relied exclusively on objective expected values. Do mOFC neurons encode objective or subjective value? Do the paper's main results on divisive normalization still hold with subjective values that incorporate risk attitudes? To answer these critical questions conclusively, it may be necessary to analyze specific trial types where expected values of safe and risky options were identical, and where objective and subjective value coding can be distinguished.

Response: We thank the reviewer for raising the point about the difference between subjective and objective (expected) value information. The reviewer raises the two specific points about the value code in mOFC neurons; i) whether the mOFC activity reflect objective values or subjective value, and ii) whether the divisively normalized value representation can reflect the subjective component of the value predicted from the risk attitude of monkeys. To answer these two questions, the reviewer suggested that we analyze specific trial types where expected values of safe and risky options were identical, and where objective and subjective value coding can be distinguished in some way.

To distinguish objective and subjective value coding in our dataset, we

performed new analyses in two ways, parametric and non-parametric approaches. In the parametric analysis, we estimated risk-attitude of each neuronal activity as well as the divisive normalization parameters as follows. We modified our advance fractional model (i.e., divisive normalization model) by replacing the expected value (EV) terms to expected utility (EU):

$$R_l = R_{\max} \frac{\beta + EU_r}{\sigma + EU_r + EU_s}$$

$$EU_r = 0.5 EV_r^\alpha, EU_s = EV_s^\alpha$$

where α is additional free parameter that represent risk-attitude of neuronal activity. Unfortunately, if we fit the model to the activity of relative value coding neurons, the model does not fit the data reliably and many estimation errors occurred.

Instead of the parametric analysis, we performed a non-parametric analysis, focusing as suggested on specific trial types in which the expected values of risky and safe options were identical. One way to examine the subjective nature of values is illustrated in the figure below: if mOFC activity reflects the risk attitude of monkeys at the moment when they performed the behavioral task, divisively normalized value signals should display a risk-attitude related degree of curvature. Specifically, activity in these equal expected value trials would systematically depend on the monkey's risk preferences because the curvature of divisively normalized value signals would change according to risk attitude of monkeys (left and middle panels). In this case, if the activity of an mOFC neuron reflects the curvature of the utility function, neuronal activity in these trials may systematically deviate from mean activity in each payoff block (right panel). To test this possibility,

we performed the following analysis to examine whether the percentage of the risky choice and the neuronal activity in the equal expected value trials were correlated

with each other.

To determine the risk attitude of monkeys at the moment when the neuronal activity was recorded, we first estimated the percentage of risky choice across all 20 lottery pairs in a recording block (risk neutral monkeys should show 50% in our lottery choice task). Then, we estimated the behavioral-neuronal correlation to examine whether the neuronal firing rates in the equal expected values trials consistently deviated from the mean firing rates of the neuron according to the monkey's risk attitude. Under subjective value coding, neural activity should systematically deviate from the linear function as a function of risk preference of monkeys as mentioned above. We tested this hypothesis by standardizing the firing rate as z score (i.e., mean is 0, variance is one), and analyzed the firing rate in the equal expected values trials in each of four payoff block in each neuron.

We plotted the z-scored firing rate in each of four payoff block in each neurons, against percentages of risky choice when that neuron activity was recorded. As shown in the figure below (supplementary figure 6), such a systematic deviation of the neuron firing rates was observed in both EVr+EVs- and EVr-EVs+ neuron types with opposite modulation effects.

This result indicates that some subjective component of value coding exists in the divisively normalized value coding. We have now added this critical result in the Results section as follows (P13).

“Normalized value signals in mOFC neurons and risk attitudes of monkeys

Lastly, we examined whether the divisively normalized value signals observed in mOFC activity were related to other aspects of the decision making process, in particular the risk attitudes of the monkeys. We estimated the correlation coefficient between behavioral risk attitudes (percentages of risky choice when a neuron was recorded) and neuronal activity, examined in trials where the expected values of

safe and risky option were identical. Specifically, we examined whether firing rates in the equal expected values trials consistently deviated from the mean firing rates of the neuron according to the monkey's risk attitude; under subjective value coding, neural activity would systematically deviate from a linear function as a function of risk preference of monkeys. We found a slight correlation between neuronal activity and percentages of risky choices with opposite signs of the effects among EVr+EVs- and EVr-EVs+ types (Supplementary Fig. 6). Thus, divisively normalized value signals in mOFC were somewhat related to the risk-attitude of monkeys.”

Since this point is important to understand the characteristics of orbitofrontal neurons in value-based decisions, we have also added the following discussion text (P14, bottom):

“Normalized value signals and choice behavior

How are divisively normalized value signals related to the monkey's choice behavior? This still remains an open question, but one possible explanation is that divisive normalization, which yields decreased neural value sensitivity with increases in total values, would yield decreased sensitivity to values in behavior, known as the subjective value or utility. Recent works in economic decision making studies have hypothesized that neuronal activity is linearly correlated with subjective value functions, an approach successfully examined in human imaging^{21, 49} and monkey electrophysiology^{50, 51} experiments. Our results suggest that the divisively normalized value signals in mOFC were at least related to the risk-attitudes observed in corresponding monkey behavior. However, the precise relationship between normalized value coding in mOFC and behaviorally-derived subjective values remain unknown, and further experimental and theoretical work will be required to link behavioral and neural observations for relative values coding.”

2. I was also surprised that canonical signals observed in (more lateral) OFC during choices were not explicitly tested, including offer value, chosen value and risky choice (Raghuraman and Padoa-Schioppa, *J Neurosci*, 2014). For comparison with existing frameworks on OFC coding, it is important to know (i) how strongly the presently reported relative expected value regressor was

correlated with the alternative explanatory variables offer value of risky or safe option, chosen value, and risky choice (the latter being a dummy variable indicating whether or not the risky option was chosen); and (ii) whether relative expected value indeed provides a better account of mOFC responses than these alternatives. I suggest performing supplementary regression analyses and goodness-of-fit comparisons to address these issues.

Response: As the reviewer correctly points out, a number of choice and value-related variables have been identified in lateral OFC. We agree that it is important to determine the potential explanatory power of these other explanatory variables, such as offer values, chosen values, and risky choices, and whether, these variables can explain the relative value modulation identified in the original manuscript. To address this point, we performed additional supplementary regression analyses to compare goodness of fit.

First, we evaluated how strongly the values derived from the different alternative value models (fractional, value difference, range normalization) were correlated with the explanatory variables noted by the reviewer: offer values of risky or safe options, chosen values, and risky choice. As shown in the supplementary table 1, the value signals in these alternative models varied, with generally moderate levels of correlation. Note that the primary model of interest that we fit to the neural data – the advanced fractional model – does not generate trial-by-trial values to compare to the other variables, as it is a nonlinear function with two free parameters (they are fitted to individual neurons). Thus, we performed a direct model comparison by estimating the correlation coefficients as suggested by the reviewer. We note here that values derived from the model most closely related to the normalization model – the fractional value model – were moderately correlated with offer values, chosen value, and risky choice (0.18-0.58).

“Supplementary Table 1. Correlation coefficients among relative expected values and other possible explanatory variables.

	EVr	EVs	EV chosen	Risky choice
Fractional value	0.58	0.18	0.38	0.56
Value difference	0.53	0.00	0.29	0.68
Range normalized value	0.80	0.41	0.63	0.59

EVr: expected values of risky options, EVs: expected values of safe options, EV chosen: expected values of chosen options, Risky choice: choice of risky option defined as a dummy variable indicating whether or not the risky option was chosen. Fractional value: $EV_r/(EV_r + EV_s)$, Value difference: $EV_r - EV_s$, Range normalized value: $(EV_r - V_{min})/(V_{max} - V_{min})$. See methods for the definition of the relative value models. ”

We have added this table to the supplementary information and also added the following description in the Methods section (P28, middle):

“ To evaluate the relationship between our primary relative value models and other known characteristics of OFC value representations, we calculated correlation coefficients between the relative expected values (derived from the fractional model, difference model, and range model) and other possible known explanatory variables ⁵⁰, such as the expected values of risky options, expected values of safe options, expected values of chosen options, and choice of risky options (Supplementary Table. 1). Note that these relative expected values were defined with no-free parameters, since the estimated free parameters mentioned above were different neuron by neuron.”

As suggested by the reviewer in his the second part of the comment, we also conducted a formal model comparison between the normalization model and these other explanatory variables to determine which model provides the best account of OFC neural activity. Specifically, we performed supplementary regression analyses and compared the AIC among those possible models. The results show that the advanced fractional model (i.e., divisive normalization model) was better able to explain mOFC activity compared to all of those alternative variables (See right figure). This figure plots the AIC differences between the advanced fractional model (M1) and other models, such as the expected values of risky options (M5), the expected values of safe options (M6), the expected values of chosen options (M7), and the choice of risky option (M8). The advanced fractional model outperformed all other models in explaining neural responses.

We have added this result in the current manuscript as a part of Figure 4c and describe the findings in the Results section. We have also added the detail of the models in the methods section as follows.

Results (P9):

“We also confirmed that the advanced fractional model was better than other potential alternative models, including ones representing the expected values of risky options, expected values of safe options, expected values of chosen options, and the choice of risky options, as well as a null model and an artificial model (Fig. 4c, $n = 81$, One sample t test, $df = 80$; M1-M5, $P < 0.001$, $t = -8.71$; M1-M6, $P < 0.001$, $t = -7.76$; M1-M7, $P < 0.001$, $t = -10.2$; M1-M8, $P = 0.009$, $t = -2.68$; M1-M9, $P < 0.001$, $t = -7.09$; M1-M10, $P < 0.001$, $t = -6.96$).”

Methods (P26 bottom to P27 top):

“Other possible alternative models

5. *Expected values of risky options.* In the model, neuronal response, R_1 , is a simple linear function of the expected values of risky options:

$$R_1 = a \text{EV}_r + b \quad (8)$$

a determines the magnitude of neural response to the expected values of risky options and b is the baseline firing rate. a and b were free parameters.

6. *Expected values of safe options.* In the model, neuronal response, R_1 , is a simple linear function of the expected values of safe options:

$$R_1 = a \text{EV}_s + b \quad (9)$$

a determines the magnitude of neural response to the expected values of safe options and b is the baseline firing rate. a and b were free parameters.

7. *Expected values of chosen options.* In the model, neuronal response, R_1 , is a simple linear function of the expected values of options monkeys chosen in the current trials:

$$R_1 = a \text{EV}_{\text{chosen}} + b \quad (10)$$

a determines the magnitude of neural response to the expected values of chosen options (EV chosen) and b is the baseline firing rate. a and b were free parameters.

8. *Choice of risky options.* In the model, neuronal response, R_1 , is a simple

function of whether monkeys chose risky option or not in the current trials (RiskyCho):

$$R_1 = a \text{ RiskyCho} + b \quad (11)$$

Where RiskyCho took scalar values (1, 0) in the trials if monkey chose risky and safe options, respectively. a determines the magnitude of neural response to the choice of risky option and b is the baseline firing rate. a and b were free parameters.”

3. To fully interpret differences in neuronal coding between free and forced tasks, we require more detailed description of the animals' behavior. For example, were there differences in performance (percentage of correct trials) and reaction times that might indicate different levels of motivation between tasks? If such differences exist, could they partly explain the diminished divisive normalization in forced trials? In particular, to test reward expectation and related motivation, did reaction times correlate with upcoming reward magnitudes in both tasks, and if so were these correlations similar between tasks? Similar correlation could strengthen the authors' conclusion that weaker divisive normalization in forced trials is related to the absence of choice requirement (rather than other factors such as motivation).

Response: We agree with the reviewer that it is important to rule out other potential explanations for the difference in neural responses between free and forced trials. We have performed two additional sets of analysis that show that differences in performance or RTs between free and forced trials do not explain the neural results. First, as suggested by the reviewer, we examined whether percentages of correct trials and saccadic reaction times were different between free and forced choice trials. Second, we directly examined the explanatory power of RT and performance in describing the neural data, via model comparison. Both of these approaches are detailed below.

As shown in the Supplementary Figure 1b in the current manuscript, saccadic reaction times related to reward magnitudes differed between task types, but the differences were not consistent between monkeys. In monkey HU, reaction times in free choice trials depended on the expected values of risky option. In forced choice trials, the effect of the risky option expected values decreased, but overall reaction times were shorter than in free choice trials. In contrast, in monkey DE, reaction

times in free choice did not depend on the expected values of the risky option. Furthermore, the reaction times were longer in forced choice trials compared to free choice trials. Thus, changes in reaction times could not consistently explain the diminished divisive normalization in forced trials. Note that saccadic reaction times were generally very fast since the timing of the GO signal was predictable in our task, with a fixed delay before the signal.

Like reaction times, performance did not display a consistent difference between free and forced trials between monkeys. As shown in Supplementary Figure 1a, the percentages of correct trials did not show a consistent free-forced pattern between monkeys. In monkey HU, there was no significant difference in performance between the free and forced choice trials. In contrast, in monkey DE the percentages of the correct trials decreased in the forced choice trials compared to the free choice trials. Together, the variable free-forced differences in both saccadic reaction times and percent correct suggest that motivation differences cannot consistently explain the diminished normalized value signals in the forced choice trials. We now describe these aspects of behavior in the revised manuscript as follows:

Result (P5 bottom):

“Behavioral measures, such as percent correct trials and saccade reaction time, were not consistently related to expected values between monkeys (Supplementary Fig. 1), suggesting that potential confounding factors such as motivation or attention did not vary between conditions.”

Supplementary Figure 1:

Supplementary figure 1. Effect of the expected values of risky and safe options on behavioral measures of monkeys.

(a) Plots of the percent correct trials of two monkeys in twenty lottery pairs during free and forced choice trials. Mean and s.e.m. are plotted. Statistical differences in each of free and forced choice trials were tested by using One-way ANOVA (Monkey HU: Free choice, $n = 999$, $df = 19$, $P < 0.001$, $F = 3.17$; Forced choice, $n = 1011$, $df = 19$, $P = 0.006$, $F = 2.02$; Monkey DE: Free choice, $n = 970$, $df = 19$, $P = 0.95$, $F = 0.54$; Forced choice, $n = 967$, $df = 19$, $P = 0.60$, $F = 0.89$). Statistical differences between free and forced choice trials were tested by using Two-way ANOVA (Monkey HU: $n = 2010$, $df = 1$, $P = 0.20$, $F = 1.62$; Monkey DE: $n = 1927$, $df = 1$, $P = 0.008$, $F = 7.00$). (b) Plots of the saccadic reaction times of two monkeys in twenty lottery pairs during free and forced choice trials. Mean and s.e.m. are plotted. Statistical differences in each of free and forced choice trials were tested by using One-way ANOVA (Monkey HU: Free choice, $n = 9979$, $df = 19$, $P < 0.001$, $F = 15.3$; Forced choice, $n = 6036$, $df = 19$, $P < 0.001$, $F = 8.60$; Monkey DE: Free choice, $n = 9604$, $df = 19$, $P < 0.001$, $F = 5.31$; Forced choice, $n = 5787$, $df = 19$, $P < 0.001$, $F = 24.4$). Statistical differences between free and forced choice trials were tested by using Two-way ANOVA (Monkey HU: $n = 16015$, $df = 1$, $P < 0.001$, $F = 19.5$; Monkey DE: $n = 15391$, $df = 1$, $P < 0.001$, $F = 137.5$). Note that mean reaction time was short because monkeys can predict the onset time of the visual signals to move eyes.”

Please note that since we have added this new figure as Supplementary Figure 1, the numbering of other supplementary figures has changed accordingly.

In addition to examining RT and performance patterns as suggested by the reviewer, we also directly test whether RT and performance can explain the neural activity differences between free and forced choice trials. Although these behavioral parameters did not show consistent free-forced patterns between monkeys, we nevertheless examined whether the percentage of correct trials and reaction times could explain the activity differences between free and forced choice trials.

To do so, we compared the performance of the following three different models - one that denotes free-forced context and two that incorporate RT or performance data - using AIC:

11. Advanced fractional model + behavioral context as dummy variables (1 for free

and 0 for forced choice trials).

12. Advanced fractional model + percent correct
13. Advanced fractional model + reaction times

As shown in the right figure, the first model – which serves to identify a block difference between free and forced choice trials – explained the neural data better than the models which included behavioral measures.

We now include a description of this result and analysis in the Results and Methods sections, and have also added the accompanying figure as Figure 7g.

Results (P12, bottom):

“In addition, the activity difference between free and forced choice trials was not better explained by behavioral measures, such as percent correct trials or saccadic reaction times (Fig. 7g, $n = 81$, One sample t test, $df = 80$; M11-M12, $P = 0.004$, $t = -2.94$; M11-M13, $P < 0.001$, $t = -3.87$).”

Methods (P30, bottom to P31 top):

“Model fit including behavioral measures as percent correct trials and reaction time

To examine whether the activity difference between free and forced choice trials could be explained by differences in state (i.e. motivation or attention) rather than differences in context, we fitted the following three modified versions of advanced fractional models. The models were simultaneously fit to both free and forced choice trial data.

$$11. \quad R_1 = R_{\max} \frac{\beta + EV_1}{\sigma + EV_1 + EV_2} + a \text{ Context} \quad (14)$$

$$12. \quad R_1 = R_{\max} \frac{\beta + EV_1}{\sigma + EV_1 + EV_2} + a \text{ Percent correct} \quad (15)$$

$$13. \quad R_1 = R_{\max} \frac{\beta + EV_1}{\sigma + EV_1 + EV_2} + a \text{ RT} \quad (16)$$

Where Context took scalar values (1, 0) in the free and forced choice trials, respectively. Percent correct was the percentages of the correct trials estimated in each of twenty lottery pairs in a given neuronal recording period. RT was the

saccadic reaction time after the central fixation target disappeared. R_{max} , β , σ and a are free parameters. We compared AIC to define which model best explained the activity difference between free and forced choice trials.”

4. The hypothesis that guides the present study is that divisive normalization is a canonical computation implemented by cortical neurons. Accordingly, the authors test the extent to which mOFC responses can be described by the formal divisive normalization model by fitting the relevant parameters to individual neuron’s activities. This approach contrasts with other approaches whereby model parameters for reward value functions are derived from fits to observed choices, and then these fitted reward values are used as regressors for neuronal activity (e.g. some of the temporal discounting and reinforcement learning studies cited in the manuscript). This latter approach has the advantage that neuronal value signals are demonstrably related to the observed behavior, as the model parameters were fitted to behavior. I wonder whether the authors can find a way to relate the presently observed divisively normalized expected value signals to the monkeys’ behavioral choices. At least it would be helpful to discuss the behavioral relevance of these signals and explain the advantages conferred by divisively normalized values for adaptive decision-making.

Response: The reviewer raises a very important point: what is the relationship between the normalization model for value coding and decision behavior? As the reviewer suggests, one approach (taken commonly in the RL and discounting literatures) is to examine whether neural responses are related to value functions – in this case, normalized value functions - estimated from the monkey’s choice behavior. For this purpose, we formalized the following mathematical model to explain choice behavior and compared estimated parameters between neuronal and behavioral ones. Unfortunately, modeling divisive normalization function based on the monkey’s behavior seemed not to be achieved well as we described below.

First, we fitted the divisive normalization model to monkey choice behavior in the trials when neuronal activity was recorded (data include a maximum 200 trials for free choice trials). We used the following equations to estimate behavioral normalization parameters:

$$DV_{risky} = R_{max} \frac{\beta + EV_r}{\sigma + EV_r + EV_s}$$

$$DV_{safe} = R_{max} \frac{\beta + EV_s}{\sigma + EV_s + EV_r}$$

$$P_{chooses}_{risky} = \frac{1}{1 + e^{-(DV_{risky} - DV_{safe}) * b}}$$

Where DV_{risky} and DV_{safe} is the divisively normalized value for risky and safe options, respectively. EV_r and EV_s are the expected values of risky and safe options, respectively. These value differences were linked to the percentage of risky choice ($P_{chooses}_{risky}$) via logistic regression. R_{max} , β , σ and b are free parameters. We also estimated divisive normalization parameters of the recorded neuron as in the manuscript, and compared these neural and behavioral parameters by scatter plot as shown below.

These results showed that the divisive normalization parameters failed to show correlation between behavioral and neuronal parameters. More precisely, a lot of the points showed similar values for the parameters estimated from behavior. This might be because the behavioral model used here did not evaluate block by block behavioral changes of monkeys precisely. Other possible reasons could exist. First, we only recorded one neuron in each session, but as our data show there is a wide diversity of normalization profiles across neurons, suggesting that there is heterogeneity in the way individual neurons normalize. This suggests that if mOFC is related to behavior, perhaps behaviorally-relevant value coding results from an averaging across all these neurons. We examined only one neuron at a time, and therefore, we would not be able to establish a consistent relationship between neural and behavioral normalization parameters because of the diversity in neural normalization across neurons. There are other reasons we might not see a significant correlation - perhaps mOFC does show normalized value but doesn't have a direct role in choice, or perhaps we don't have the right model for how to fit

normalized value from decision behavior.

Therefore, we have added text to the Discussion about what is the adequate model to explain normalization in value based decision making as follows (P16, middle):

“Normalized value signals and choice behavior

How are divisively normalized value signals related to the monkey’s choice behavior? This still remains an open question, but one possible explanation is that divisive normalization, which yields decreased neural value sensitivity with increases in total values, would yield decreased sensitivity to values in behavior, known as the subjective value or utility. Recent work in economic decision making studies hypothesized that neuronal activity is linearly correlated with subjective value functions, an approach successfully examined in human imaging^{21, 49} and monkey electrophysiology^{50, 51} experiments. Our results suggest that the divisively normalized value signals in mOFC were at least related to the risk-attitudes observed in corresponding monkey behavior. However, the precise relationship between normalized value coding in mOFC and behaviorally-derived subjective values remain unknown, and further experimental and theoretical work will be required to link behavioral and neural observations for relative values coding.”

Minor concerns

5. From the description it seems that the ratio of safe to risky choices in the forced task was not matched to the animals’ specific preferences during the free task. Could this difference have affected the neuronal coding?

Response: The reviewer is correct that the ratio of safe to risky choices in the forced task was not matched to the animals’ specific preferences during the free choice trials. The forced choice trials were composed of 36 trials in each payoff block, and we allocated the monkey’s instructed choices to equally sample all of the five risky options and one safe option 6 times each (6 repetition time 6 conditions, 36 trials). Since the percentage of risky choice was fixed in the forced trials, this yields a difference in the frequency of the risky choices between free and forced choice trials in each lottery pair. As pointed out by the reviewer, an important question is whether this difference could be driving the observed difference in

neural coding between free and forced trials. However, the new analyses described above (in the reply to point No. 2 of the first reviewer) show that monkey's choice of risky option cannot explain the mOFC activity well. Thus, the allocation of risky choice in the forced choice trials seemed not to explain the activity changes in the forced choice trials compared to free choice trials.

6. I do not understand what was tested in lines 208-212 to rule out an adaptation process. Please provide a more detailed description.

Response: We apologize for the lack of clarity in our previous manuscript, and have revised the manuscript to better describe the analyses. Our intent was to determine if the differences between forced and free choice activity arise due to a slow change in activity (such as adaptation) after a change in reward structure at the onset of a new block. If there exists an adaptation process to the new reward values that occur at the start of a block, activity modulation in the forced choice trials would gradually change. Since forced choice trials always occur first, forced choice trials would represent the start of the adaptation process while free choice trials would represent the end of the adaptation process.

To examine if such a process could explain the observed free-forced differences, we examined how modulation varied as a function of progress through both free and forced choice trial blocks. In contrast to an adaptation-like process, weak activity modulation during the forced choice trials was stable across all forced choice trials and activity modulation only changed sharply after the start of the free choice trials (Figure 7b in the current manuscript). In this figure, we showed that stable regression coefficients were observed after the payoff block change in the forced choice trials, but sharp change of the regression coefficients were observed after the start of free choice trials. We improved our description as follows (P11, middle).

From: "While the forced choice trials were presented to the monkeys at the beginning of PBs, weak modulation in the forced choice trials was not due to an adaptation process, as might be postulated to occur in adjacent area IOFC(Fig. 6b, One-way ANOVA: forced choice trials, $n = 486$ ($81 \times 3 \times 2$), $P = 0.75$, $F = 0.29$, $df_{212} = (2, 483)$; free choice trials, $n = 648$ ($81 \times 4 \times 2$), $P = 0.35$, $F = 1.09$, $df = (3, 644)$)."'

To: “While the forced choice trials were presented to the monkeys at the beginning of PBs, weak modulation in the forced choice trials was not due to an adaptation process, as might be postulated to occur in adjacent area IOFC^{11, 12}. The weak modulation in the forced choice trials were maintained throughout forced choice trials (Fig. 7b, One-way ANOVA: forced choice trials, $n = 486$ ($81 \times 3 \times 2$), $P = 0.75$, $F = 0.29$, $df = (2, 483)$). Stronger modulation appeared only after the start of free choice trials (paired t test, $P < 0.001$, $t = 3.66$, $df = 161$, the last 12 forced choice trials vs. the first 12 free choice trials) and was maintained through a payoff block (One-way ANOVA: free choice trials, $n = 648$ ($81 \times 4 \times 2$), $P = 0.35$, $F = 1.09$, $df = (3, 644)$).”

7. So far, few studies have directly compared neuronal coding of valuation and decision variables in free choice and forced tasks. The demonstration of diminished relative valuation in forced trials is a clear strength of the present study. A few recent papers on amygdala neurons in economic choice reported similar phenomena, including relative specificity of value signals to free choice (Hernadi et al., *Nature Neuro*, 2015) and diminished neuronal adaptation during forced trials (Grabenhorst et al., *eLife*, 2016), which may inform the discussion in lines 273-283.

Response: As the reviewer suggests, we have improved the Discussion as follows (P15, middle):

“Enhanced value signals during free choice have also been found in the activity of monkey amygdala neurons^{41, 42}, which is connected to the orbitofrontal cortex.”

8. It would be helpful to report the amount of explained variance by the value regressor for the example neuron.

Response: As the reviewer suggests, we have added the description of explained variance as follows (P8, bottom):

“percent variance explained: trial-based, 13.5%; mean responses-based, 46%”

9. The plot in Fig. 6b showing transition in neuronal coding from forced to free trials is particularly striking, as it clearly indicates the rapidity of change in mOFC value coding. Could the authors quantify this difference statistically, e.g. demonstrating significant change in coefficients from last forced to first free trial block?

Response: As the reviewer suggests, we applied a t-test between the last forced choice 12-trial block and the first free choice 12-trial block. We have added the following statistical description in the result section (P11, L13-14):

“Stronger modulation appeared only after the start of free choice trials (paired *t* test, $P < 0.001$, $t = 3.66$, $df = 161$, the last 12 forced choice trials vs. the first 12 free choice trials)”

Reviewer #2 (Remarks to the Author):

Yamada H et al., Free choice shapes normalized value signals in medial orbitofrontal cortex

The authors analyze data recorded in mOFC in a task where monkeys chose between sure and probabilistic outcomes. They found that animals chose the probabilistic outcome as its expected value increased. When they examined neural activity they found that it scaled relative to the other offers available in a block of trials. They fit several models to the neural activity and found that the advanced fractional model, a divisive normalization model, fit the data best at the population level. In forced choice trials this was also true, although the scaling was less pronounced.

The coding strategies used in areas that represent choice values have been explored in much less detail than the coding strategies used in primary sensory areas. This paper begins to make progress on this question by identifying aspects of the coding in mOFC. The paper is clearly written and straightforward. A few details were unclear or I missed them.

Response: We thank the reviewer for their favorable comments, and have addressed their concerns as detailed below.

1. What was the likelihood model? Gaussian? The AIC differences seemed relatively small. Were these based on single trials, or averages across trials for each condition? Also, in Fig. 4b, is this across task epochs? Or does each neuron contribute just once to this? How frequently were neurons significant in more than one epoch? Were the 81 neurons that were analyzed neurons that were significant in all epochs, or at least one?

Response: We appreciated the reviewer's careful reading of our approach, and apologize for any lack of clarity in our previous descriptions. Below, we detail our responses to each point and accompanying revisions to the manuscript.

What was the likelihood model? Gaussian?

Response: The likelihood of the model was Gaussian. In practice, we estimated the likelihood of the linear and non-linear models using the software “R”. We used nls() function to fit the model, in which nonlinear least squares are used to estimate the free parameters of the models. Maximum log likelihood was estimated after the fits with an underlying assumption that the least squared values are linearly summated. We added the following description to the method section (P28, bottom):

“In this function, a set of parameters that minimize non-linear least squared values were estimated.”

The AIC differences seemed relatively small. Were these based on single trials, or averages across trials for each condition?

Response: AIC was estimated based on single trials. We fitted the models to the single neuron activity by estimating the AIC based on single trials. Since trial-based data generally exhibit large variability in firing rates due to the nature of cortical spiking, it is not unexpected to find relatively small AIC differences between models. We apologize that this point was not made clearly enough in the original manuscript, and have added the following description in the Methods section (P28, bottom):

“Best fit parameters were estimated in each epoch of the activity of the neuron based on single trial firing rates.”

in Fig. 4b, is this across task epochs? Or does each neuron contribute just once to this? How frequently were neurons significant in more than one epoch? Were the 81 neurons that were analyzed neurons that were significant in all epochs, or at least one?

Response: We apologize for the confusion, and clarify the details below. This example showed the activity during a single task epoch, which is the cue period. In this example, the cue period activity was used for model comparisons in which we estimated AIC values in each of four models described in the figure 4a.

In our population analyses shown in Figure 4b, all epochs that showed relative value coding were used. mOFC neural activity encoding relative value was detected in three task epochs: cue period (1.0 s window after cue onset); saccade period (1.0 s window after saccade onset); and feedback period (1.0 s window after feedback onset). Relative value coding was identified in 81 activity epochs from 101 neurons, while the number of possible epochs were 303 (101 neurons times 3 epochs). If we summarize the activity modulation in each neuron, relative value modulation was found in 48 of 101 neurons in at least one of three task epochs, and the average number of task epochs per neuron showing relative value modulation was 1.67. Since our description was not clear in the original manuscript, we have improved the description as follows (P7, L7-11):

“In total, 27 percent (81/303) of the task periods showed activity modulation by the relative value of options, and 48 neurons exhibited relative value coding in at least one of the three task epochs. These 81 relative value signals were used in further analyses to test in greater detail how the value signals are normalized.”

In the legend of Figure 4b, we described as follows (P41, L7-9)

“Mean and s.e.m were estimated for the 81 activity of neurons that showed the relative value coding.”

2. Does the divisive normalization model perform better than a model which floats the intercept and slope for each block?

Response: To answer the reviewer’s query, we examined a model which has four independent intercept parameters and four independent linear regression slopes corresponding to the four payoff block of trials. Then, we compared AIC values by calculating the AIC difference between this artificial model and the divisive normalization model (M1). As shown in the Figure 4C and here, this additional alternative model (M10) performed worse than the advanced fractional model in explaining neural responses. We now describe this results in the Results and Method sections as follows:

Results (P9):

“We also confirmed that the advanced fractional model was better than other potential alternative models, including ones representing the expected values of risky options, expected values of safe options, expected values of chosen options, and the choice of risky options, as well as a null model and an artificial model (Fig. 4c, $n = 81$, One sample t test, $df = 80$; M1-M5, $P < 0.001$, $t = -8.71$; M1-M6, $P < 0.001$, $t = -7.76$; M1-M7, $P < 0.001$, $t = -10.2$; M1-M8, $P = 0.009$, $t = -2.68$; M1-M9, $P < 0.001$, $t = -7.09$; M1-M10, $P < 0.001$, $t = -6.96$).”

Methods (P27-28):

“10. An artificial model. In the model, neuronal response, R_1 , is a function of the expected values of risky options in each payoff block:

$$R_1 = a_1 EV_r + b_1 + a_2 EV_r + b_2 + a_3 EV_r + b_3 + a_4 EV_r + b_4 \quad (13)$$

a_1 - a_4 determine the magnitude of neural response to the expected values of risky options in the payoff block number 1 to 4, respectively. b_1 - b_4 are the baseline firing rate in the payoff block number 1 to 4, respectively. a_1 - a_4 and b_1 - b_4 are free parameters.”

3. Is the neuron in Fig. 4 the best typical example?

Response: This neuron is not the best example, but instead ranked 16th out of 81 relative value-coding activity of the mOFC neurons in percent variance explained estimated based on the single trials. Thus, this is not the best example, but was positioned at the 20th percentile.

Reviewer #3 (Remarks to the Author):

The manuscript reports on new experiments and analysis, addressing the role of medial Orbitofrontal Cortex (mOFC) neurons in coding value signals. The study is a nice follow up on previous work by the same lab on 'relative' value coding and divisive normalization, and reports two main findings: 1) mOFC neurons can be described by divisive normalization, in which the expected value of a choice option is normalized by the expected value of an alternative option (defined as free-choice context); 2) the normalization seems abolished when the alternative option has zero value (defined as forced-choice context).

Overall the manuscript is interesting and well written. The problem is clearly stated and placed in the context of relevant literature; the approach is well justified; and the findings are original to the best of my knowledge, and would be of interest to others in the neuroscience community. However, my main concern is that the results are not convincing and, in my opinion, do not support quantitatively the proposed conclusion, as detailed below. I have also a couple of minor comments on the discussion of related literature.

Response: We greatly appreciate the reviewer's insightful comments, which raised important points that allowed us to improve our analyses and conclusions and strengthen our manuscript dramatically. Below we address each of the comments in turn.

Major

To support the main finding of context dependent normalization in mOFC, the authors propose a model of single-neuron firing rate based on divisive normalization (equation on line 147), fit it to data and compare it to alternative models. I find a number of issues with the analyses and the results reported.

1) Poor fits.

Primarily, the model fits are poor even for the example cell (Figure 4), and a disaster across the population (in Supplementary Figure 3, the variance explained is <10%!). The authors compare models using relative AIC scores (more on this below), but they do not report quantities that would more clearly

capture fit quality. One obvious choice is R-square or variance explained, which is relegated to just one Supplementary figure where it shows that the model only captures <10% of the variance of the data. This may be in line with other work in OFC (of which I am no expert) and if so it would be useful to state it; but, even then, I find it hard to build a case that these neurons encode relative value in a context-dependent manner when relying on a model that misses more than 90% of the response.

I would suggest the authors dig into the qualitative aspects of neural responses that the are missed by the model, and then explore variants of divisive normalization that could improve model fits. For instance, the model seems to miss the strong saturation at moderate and high EVr, and perhaps adding an exponent to the denominator could correct that.

Response: We greatly appreciate the reviewer's points regarding evaluating model fit performance. As the reviewer pointed out, we did not clearly describe quantities to capture the model fit quality. Furthermore, we did not specify the trial-by-trial resolution of the explained variance in the original manuscript. As a result, the previous version of the manuscript did not adequately convey the fit quality of the normalization model. Below, we (1) explain our previous explained variance measures and the reason why the values appeared to be low, as well as (2) recalculate explained variance in line with the previous procedures used in the analysis of OFC responses, as pointed out by the reviewer.

In the original manuscript, estimation of percent variance explained was based on single trials in each neuron, and unfortunately we did not describe this critical point in sufficient detail. This issue was also pointed out by the second reviewer (comment number 1). We have now corrected the manuscript by explicitly stating that the model fit was based on single trials:.

Methods (P28, L20-21):

“Best fit parameters were estimated in each epoch of the activity of the neuron based on single trial firing rates.”

Since the single trial-based model fit contains trial-by-trial variability in firing rates, the percentage of variance explained is not high compared to previous OFC studies in which R-squared values were estimated based on the mean response of a neuron in each trial condition (e.g. Raghuraman and Padoa-Schioppa, 2014, J Neurosci).

To directly compare the performance of our model fit to the previous OFC study, we now estimate the percent variance explained based on mean response in each lottery pair in the revised manuscript. Estimating percent variance explained based on the mean responses results in larger measures (e.g. 46% in the example activity in Fig. 4a) because trial-by-trial fluctuations spiking variance – which is not addressed by mean rate models – are removed. Indeed, this mean response-based value is close to the value observed in the previous OFC study, in which the % variance explained were estimated based on mean response in each condition. On average, percent variance explained estimated based on mean responses was about 40% and significantly higher than all other alternative models, as shown in Figure 5a of the current manuscript (and detailed below). It was sure that percent variance explained estimated based on mean responses were larger than percent variance explained estimated based on single trial (Figure 5b).

To make this important point clear, we have improved our description of how we evaluated model performance and now include the estimation of percent variance explained based on single trials as well as mean response in each lottery pair. In addition, we have added a new Figure 5 and additional description in the Results and Methods sections as follows:

Figure 5a and b

Figure 5. model performance for relative value coding.

(a) Plots of the percent variance explained by the normalization model for the mean response data in twenty lottery pairs. (b) same as a, but for the single trial-based data.

Results (P9 to P10):

“To evaluate the performance of the model, we estimated percentages of variance explained (see Methods). The divisive normalization model performed well compared to the other three relative value models (Fig. 5), as 40% of the variance was explained by the advanced fractional model in the mean response-based estimation in 20 lottery pairs (Fig. 5a, $n = 81$, paired t test, $df = 80$; M1 vs. M2, $P < 0.001$, $t = 8.38$; M1 vs. M3 EVs, $P < 0.001$, $t = 6.54$; M1 vs. M4, $P < 0.001$, $t = 7.65$). Similar results were obtained when the percent variance explained was estimated based on single trial data (Fig. 5b, $n = 81$, paired t test, $df = 80$; M1 vs. M2, $P < 0.001$, $t = 5.87$; M1 vs. M3 EVs, $P < 0.001$, $t = 4.94$; M1 vs. M4, $P < 0.001$, $t = 5.55$), though as expected the single trial-based percent variance explained was lower than the mean response-based measure due to trial by trial variability in the neural activity.”

Methods (P29):

“Evaluation of model performance

To evaluate model performance, we estimated the percentages of variance explained, which is defined as one minus percentage of the residual variances out of total variances. The percent variance explained in each neuron was estimated based on either single trial data or mean responses data (segregated by the twenty lottery pair conditions).”

We also realized that this estimate is analogous to the measure known as explainable variance used in the vision studies, which is usually defined as percent variance explained based on the mean responses to each visual stimuli. We now explain this similarity between our measure and explainable variance in the Methods section.

Methods (P29):

“The mean response-based percent variance explained is similar in principle to explainable variance⁵⁵.”

In summary for our reply to this specific point, our previous manuscript did not assess model performance in sufficient detail, particularly in a way to allow comparison to previous papers. The results of our new analysis - variance explained estimated based on the mean-response in twenty lottery pairs - clearly show that the advanced fractional model performs well in characterizing mOFC neural responses. We also realized that this estimate is analogous to the measure known as explainable variance used in the vision studies, which is usually defined as percent variance explained based on the mean responses to each visual stimuli.

Also a couple of technical suggestions to improve fit quality: I would normalize the rates by their across-trial variances (eg see Cavanaugh et al 2001 J Neurophys), or consider some other noise model that is more accurate for cortical neural activity – if I understand correctly the Methods, the current fits assume additive noise of constant amplitude (not stated explicitly, but seems implied). Second, rather than percent explained variance, the authors should report percent ‘explainable’ variance, ie. normalized to a noise ceiling imposed by the limited number of measurements. Related: for completeness, I would suggest reporting the number of repeats per condition that are used to measure firing rates; this will become relevant when computing the noise ceiling.

Response: In the last paragraph of this first comment, the reviewer raised variety of technical suggestions to improve our model fits since the model performance in the previous manuscript appeared to be very poor. We thank the reviewer for these clear suggestions, which emphasized the lack of clarity in our description of our approach in the previous manuscript. Among all suggestion in the first comment, we chose the one that has been used in the previous OFC study, percent variance explained based on mean responses, while this estimate was analogous to percent ‘explainable’ variance. The analysis suggested by the reviewer has led us to improve our model evaluation dramatically as described above. This new analysis supports the idea that advanced fractional model was better than other possible models in characterizing the neural responses. Given the comparability between our findings and previous work in the OFC using mean response explained variance, we thought that it is better to describe our results without other suggested analyses because we would like to avoid the additional complexity in our analysis.

In this paragraph of the first comment, the reviewer also suggested to clarify

the number of repeats per condition for completeness. We now clearly state this number in the Methods section as follows.

Methods (P20, middle):

“The first 36 trials (6 repeats times 6 conditions (five risky and one safe choices)) in a block were forced choice trials. Then, 50 free choice trials (10 of each 5 type) followed.”

Lastly, these first comments about the fitting performances allowed us to improve our manuscript dramatically, and that makes our conclusion clearer. We appreciate the reviewer’s insightful comments, again.

2) Imperfect model comparison.

First, I suggest the authors use a different metric to compare models. Intuitively, AIC penalizes any parameter by the same amount, which is fine for models that are linear in the parameters (and under the assumption of Gaussian observation noise). But the normalization models are not linear in the parameters, which means that, for instance, the additional model expressivity (or complexity) offered by σ in model 1 is not the same as ‘b’ in the other models. The most straightforward choice would be cross-validated percent explainable variance (ie normalized model log likelihood). Another, more principled option, is Bayesian Model Comparison, or the ‘widely applicable information criterion’ (WAIC), as they take into account the full posterior over parameters. The choice will depend on the specifics of the data (primarily, number of data points available).

Response: The reviewer makes an excellent point regarding cross-validation vs. AIC as a metric for model comparison. As the reviewer suggests, we applied cross validation to estimate cross validated explainable variance, i.e., cross validated percent variance explained based on the mean response in each of the twenty lottery pairs.

As shown in the right figure and Figure 5c, two-fold cross validation analysis gives the same qualitative results

as our original AIC analyses regarding the performance of the models. In the training data and in the test data, percent variance explained was highest in the advanced fractional model (M1). We have added this result as Figure 5c and additional text in the Results and Methods sections as follows:

Results (P9, bottom to P10 top):

“Furthermore, cross validation demonstrated model explanatory power in test data as well as training data, with the advanced fractional model remaining the best model (Fig. 5c, test data: $n = 81$, paired t test, $df = 80$; M1 vs. M2, $P < 0.001$, $t = 5.39$; M1 vs. M3EVs, $P < 0.001$, $t = 4.55$; M1 vs. M4, $P < 0.001$, $t = 5.45$). Note that percent variance explained decreased even in the training data since the data size was half the size of the full data set.”

Methods (P29, middle):

“To validate the accuracy of estimation and model selection, we performed two-fold cross validation (i.e., half split) in each of the model fits as follows. First, we prepared training data and test data, by randomly dividing the data in half in each of twenty lottery pairs. Models were fitted to the training data and best fit parameters were estimated. By using these estimated parameters, percentages of variance explained were estimated for the test data.”

Second, as the fit quality is so poor, one is left wondering if a ‘null’ model that assumes constant response (ie just 1 free parameter) would compare better than all of the proposed models, once accounting properly for model complexity.

Response: The reviewer makes a valid point about potential pitfalls of penalizing for model complexity in model comparison. While we have shown above that the model fits are reasonable when addressing mean firing rates and that the normalization model outperforms alternative models under cross-validation, we nevertheless now also examine the suggested null model. We have added the null model - which contains just a single parameter to explain the mean firing rate – as an alternative model. As shown in the Figure 4c, the null model (M9) was clearly performs worse than the advanced fractional model in model comparison using AIC. We now describe this additional result in the revised manuscript as follows:

Results (P9, top):

“We also confirmed that the advanced fractional model was better than other potential alternative models, including ones representing the expected values of risky options, expected values of safe options, expected values of chosen options, and the choice of risky options, as well as a null model and an artificial model (Fig. 4c, $n = 81$, One sample t test, $df = 80$; M1-M5, $P < 0.001$, $t = -8.71$; M1-M6, $P < 0.001$, $t = -7.76$; M1-M7, $P < 0.001$, $t = -10.2$; M1-M8, $P = 0.009$, $t = -2.68$; M1-M9, $P < 0.001$, $t = -7.09$; M1-M10, $P < 0.001$, $t = -6.96$).”

Methods (P27, bottom):

“9. Null model. In the model, neuronal response, R_1 , is only a function of the mean firing rate:

$$R_1 = b \quad (12)$$

b determines the mean firing rate. b was free parameter.”

Minor

1) I found the statement in the first Discussion section (starting on line 258) a bit too strong. The authors suggest that “the clear distinction between mOFC and IOFC in terms of relative value coding is the dependency on the behavioral context.” In this manuscript, the authors use an experimental design that probes different contexts explicitly, and quantitative analysis based on model comparison, to conclude that there is context dependence. As far as I understand the cited reference on IOFC, in that experiment there is only one context, so it is hard to conclude so strongly that IOFC would not show context dependence, if one were to test it.

Response: As the reviewer suggests, we have toned down the discussion section and made the statement regarding the experiment clearer as follows:

(P15, middle):

“Although it is not known what type of normalization is employed in IOFC neurons, one possible distinction between mOFC and IOFC in terms of relative value coding is the dependency on the behavioral context. The free choice-specific relative value coding in mOFC (Fig. 7) may represent a key difference from IOFC neurons, where relative value signals are observed even in non-choice situations during classical

conditioning¹².”

2) Overall, I found the discussion of related work on efficient coding and normalization in sensory processing, light and rushed (please check for typos). In particular, the finding that divisive normalization operates under one context but is abolished under a different context has been shown recently in visual cortex (Coen-Cagli et al 2015 Nature Neuro) and described by a model equivalent to the one dubbed here “assumption 2” where the context term in the normalization signal is set to zero.

Response: We apologize that the relevant section of the previous Discussion was not adequately clear. The Coen-Cagli reference is particularly relevant; as the reviewer suggests, we have revised the discussion as follows:

(P17):

“Efficient coding and divisive normalization

The efficient coding hypothesis assumes that the neural code adapts efficiently to the present behavioral context, and that neurons change their firing rates in order to utilize their entire dynamic range during encoding¹³. Efficient coding requires input–output functions to use the entire response range to represent the stimulus distribution⁵². In the domain of sensory systems for perception, a large literature supports the hypothesis that normalization is employed to achieve efficient coding¹⁷. Moreover, a recent finding by Coen-Cagli et al shows that normalization processes in primary visual cortex can be flexibly gated depending on the sensory context⁵³. In contrast to the sensory domain, only a couple of direct and indirect tests have been conducted to examine the implementation of efficient coding in decision making^{11, 12, 14, 34}. Our current study highlights that value-based divisive normalization occurs in frontal decision circuits; furthermore, the modulation of this normalization by the behavioral choice context suggests that the flexible gating of contextual information occurs in both sensory and decision-related computations.”

3) An interesting prediction of the advanced fractional model is that the baseline firing rate of a neuron of the class EVr+ EVs- should be modulated by the safe

option value even in the absence of a risky option. Would it be possible to test this in the existing data?

Response:

We analyzed the baseline firing rate of the relative value coding neurons which showed relative value signals at least in one of three task epochs. We examined whether the baseline activity of these 48 neurons showed significant modulation by the expected values of safe option based on the regression analysis. As shown in the right figure, regression coefficients of the expected values of safe option showed significant deviation from zero in the EVr-EVs+ type, but did not show such modulation in the EVr+EVs- type. Thus, baseline activity may be or may not be modulated by the expected values of safe options.

4) For clarity, it would be useful to state explicitly what is the time window used to estimate the firing rates that went into the models (eg around line 161). Also, on line 205, I would remind the reader what 'regression coefficients' refer to.

Response: We described the time window for the analysis in the results sections and Methods as follows.

Results (P7 top)

“see gray lines in Fig. 1a for three task periods: cue period (1.0 s window after cue onset); saccade period (1.0 s window after saccade onset); feedback period (1.0 s window after feedback onset).”

Methods (P21, bottom)

“We analyzed neuronal activity during three task periods: cue period during 1.0 s after cue onset; saccade period during 1.0 s after saccade onset; feedback period during 1.0 s after feedback onset.”

Results (P10, start of the second paragraph)

“When the monkeys were instructed by the computer to “choose,” the relative value signals evident in the regression coefficients for the expected values of risky and safe options were weak when compared to those observed on free choice trials in the activity of the same neurons.”

Typos: line 80, ‘situation*al*’; line 293, ‘studies’ repeated twice

Response: We have corrected these items.

REVIEWERS' COMMENTS:

Reviewer #1 (Remarks to the Author):

The authors have appropriately addressed all points that had been raised in this careful and thorough revision. This is a very interesting manuscript and I recommend publication of the current version without further revision.

Reviewer #2 (Remarks to the Author):

The authors have addressed all of my concerns. I have no further comments.

Reviewer #3 (Remarks to the Author):

All my comments were addressed satisfactorily, and I find the manuscript much improved overall.